# Cross-modal autoencoder framework learns holistic representations of cardiovascular state

Adityanarayanan Radhakrishnan[1,5], Sam F. Friedman [2,5], Shaan Khurshid[2,3], Kenney Ng [4], Puneet Batra [2], Steven A. Lubitz [2,3] ✉, Anthony A. Philippakis [2] ✉ & Caroline Uhler [1,2] ✉

A fundamental challenge in diagnostics is integrating multiple modalities to develop a joint characterization of physiological state. Using the heart as a model system, we develop a cross-modal autoencoder framework for integrating distinct data modalities and constructing a holistic representation of cardiovascular state. In particular, we use our framework to construct such cross-modal representations from cardiac magnetic resonance images (MRIs), containing structural information, and electrocardiograms (ECGs), containing myoelectric information. We leverage the learned cross-modal representation to (1) improve phenotype prediction from a single, accessible phenotype such as ECGs; (2) enable imputation of hard-to-acquire cardiac MRIs from easy-to-acquire ECGs; and (3) develop a framework for performing genome-wide association studies in an unsupervised manner. Our results systematically integrate distinct diagnostic modalities into a common representation that better characterizes physiologic state.

Clinicians leverage measurements across many complementary diagnostic modalities to develop an integrated understanding of a patient's physiological state. For example, heart function can be interrogated with a variety of modalities, such as electro-cardiograms (ECGs) that provide myoelectric information (e.g. sinus rhythm, ventricular rate, etc.), and cardiac magnetic resonance images (MRIs) that provide structural information (e.g. left ventricular mass, right ventricular end-diastolic volume, etc.). By utilizing measurements across both modalities, we can gain a more holistic view of cardiovascular state than with either modality alone. The recent availability of large-scale cross-modal measurements in biobanks[1,2] provides the opportunity to develop systematic and rich representations of physiology. In particular, such cross-modal representations provide an opportunity for a broad range of downstream tasks such as (1) prediction of clinical phenotypes for diagnostics; (2) imputation of missing modalities in biomedical data; and (3) identification of genetic variants associated with a

given organ system. Using the heart as a model system, we here develop such an integrative framework and show its effectiveness in these three downstream tasks.

Multi-modal data integration is a rich field with a variety of methods developed for specific applications. A survey of multi-modal approaches is presented in ref. 3. Unlike multi-modal data integration approaches based on classical methods such as canonical correlation analysis (CCA)[4–7] or non-negative matrix factorization[8,9], our approach relies on a class of machine learning models called autoencoders. Autoencoders[10,11] are a class of generative models that serve as a standard method for learning representations from unlabelled data. These models have been successfully applied in a variety of applications including computer vision[12–14], chemistry[15], and biology[16–21]. A line of recent works utilize autoencoders to learn joint representations of multi-modal data including natural images and captions in computer vision[14,22–27], nuclear images and gene expression in biology[20], and paired clinical

[1]Massachusetts Institute of Technology, Cambridge, USA. [2]Broad Institute of MIT and Harvard, Cambridge, USA. [3]Massachusetts General Hospital, Massachusetts, USA. [4]IBM T.J. Watson Research Center, New York, USA. [5]These authors contributed equally: Adityanarayanan Radhakrishnan, Sam F. Friedman. ✉e-mail: lubitz@broadinstitute.org; aphilipp@broadinstitute.org; cuhler@mit.edu

measurements[28–31]. Indeed, autoencoders have been observed to perform competitively with other multi-modal integration methods including classical integration approaches using CCA[5–7] and generative adversarial networks[32,33]. Unlike these prior works that focus primarily on improving a specific downstream task such as phenotype prediction or modality translation through multi-modal data integration, we develop a generalized representation that improves performance on several downstream applications. We demonstrate the utility of this representation on three important biomedical tasks: in addition to phenotype prediction and multi-modal data integration and translation, we show that our cross-modal representation yields a new framework for characterizing genotype–phenotype associations. While various prior works have conducted genome-wide association studies (GWAS) to identify single nucleotide polymorphisms (SNPs) associated with cardiovascular diseases[34,35], features measured on ECGs[36,37], or features measured on cardiac MRI[38,39], these GWAS approaches have relied on labelled data derived from individual modalities. Instead, our approach can identify SNPs that affect cardiac physiology in an unsupervised and general way. Namely, rather than merely identifying SNPs that affect a single phenotype such as the QT interval, our approach identifies SNPs that generally impact phenotypes present on ECGs or cardiac MRIs.

Utilizing cardiac MRI and ECG samples from the UK Biobank[1], we develop a cross-modal autoencoder framework for building a representation of cardiovascular state (Fig. 1a). We show that these learned representations improve phenotype prediction (Fig. 1b) over supervised deep learning methods. Additionally, our cross-modal autoencoders enable generating hard-to-acquire MRIs from easy-to-acquire ECG samples, and we show that these generated MRIs capture common MRI phenotypes (Fig. 1c). We show that a GWAS on phenotype labels derived from cross-modal embeddings leads to the recovery of known genotype–phenotype associations. Importantly, our framework also allows us to perform GWAS in the absence of labelled phenotypes to identify SNPs that generally impact the cardiovascular system (Fig. 1d).

## Results

### Cross-modal autoencoder framework enables the integration of cardiovascular data modalities

To build a cross-modal representation of the cardiovascular state, we utilize autoencoders to map paired cardiovascular data modalities, i.e. 38,686 paired median 12-lead ECGs and 50 frame videos of long-axis cardiac MRIs, from the UK Biobank[1] into a common latent space. A description of the data used in this work is provided in the "Methods" subsection "Study design". Building on the traditional autoencoder framework, we train modality-specific encoders and decoders to map to and from this latent space such that the reconstructed training examples are similar to the original examples for all modalities (see Fig. 1a). Additionally, given an ECG and MRI pair for a single individual, we utilize a loss function that ensures that paired ECG and MRI samples are represented via nearby points in the latent space (i.e., using a *contrastive loss*). Importantly, while our model is trained on paired modalities, the model can be applied in settings where only one modality is present. Namely, we simply utilize the embedding given by the trained encoder for the single input modality. A description of our loss function, architecture, and training procedures is given in the "Methods" subsection "Cross-modal autoencoder architecture and training details" and Supplementary Fig. S1. As indicated in Fig. 1, the resulting representations are useful for a variety of downstream tasks including phenotype prediction, modality translation, and genetic discovery. While we mainly apply our framework to integrate two modalities (ECGs and cardiac MRIs), we demonstrate that it can also be applied to three or more modalities in Supplementary Fig. S2.

### Cross-modal representations enable improved phenotype prediction

We first demonstrate that supervised learning on cross-modal representations improves performance on phenotype prediction tasks. While our model is trained on ECG and MRI pairs, we consider the practically relevant setting in which only one modality (e.g. ECG) is available. In this case, we perform supervised learning on embeddings given by a single modality-specific encoder (Fig. 1b). For our cross-

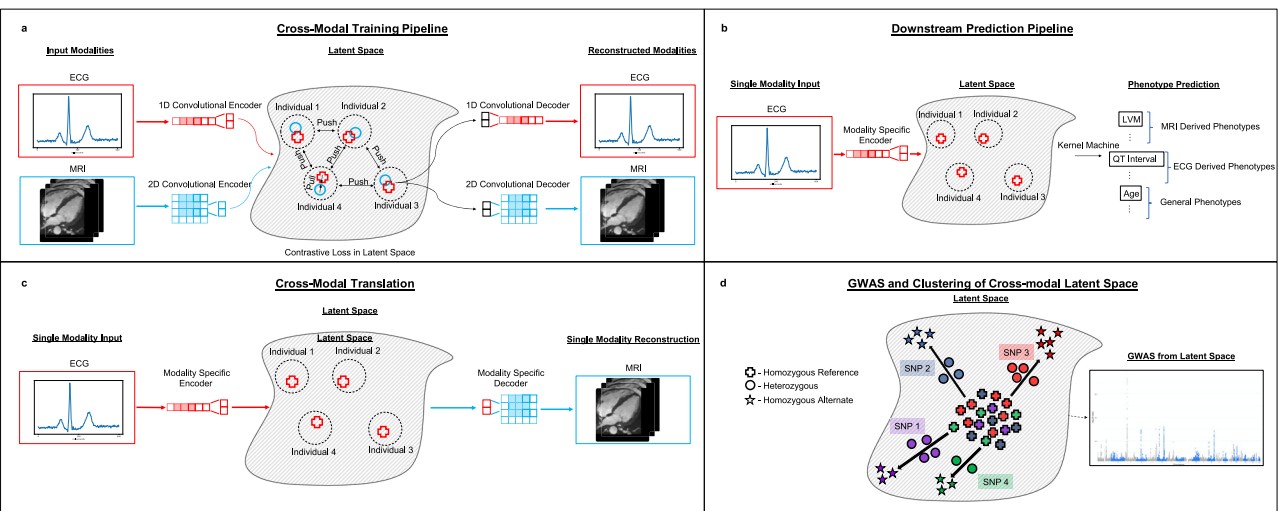

**Fig. 1 | An overview of our cross-modal autoencoder framework for integrating cardiovascular data modalities.** Our model is trained on ECG and cardiac MRI pairs from the UK Biobank. **a** A visualization of our training pipeline. Modality-specific encoders map data modalities into a shared latent space in which a contrastive loss is used to enforce the constraint that paired samples are embedded nearby and further apart from other samples. Modality specific decoders are then used to reconstruct modalities from points in the latent space. **b** Learned cross-modal representations are used for downstream phenotype prediction tasks by training a supervised learning model (e.g., a kernel machine) on the latent representations. **c** Our framework enables translation between modalities: ECGs can be translated to corresponding MRIs and vice-versa. **d** The learned cross-modal representations can be used to understand genotype-phenotype maps in the absence of labelled phenotypes by performing a GWAS in the cross-model latent space and clustering SNPs via their signatures (i.e., the vector in latent space oriented from homozygous reference to the mean of heterozygous and homozygous alternate); SNPs 1 and 4 have similar signatures in the latent space and thus similar phenotypic effects.

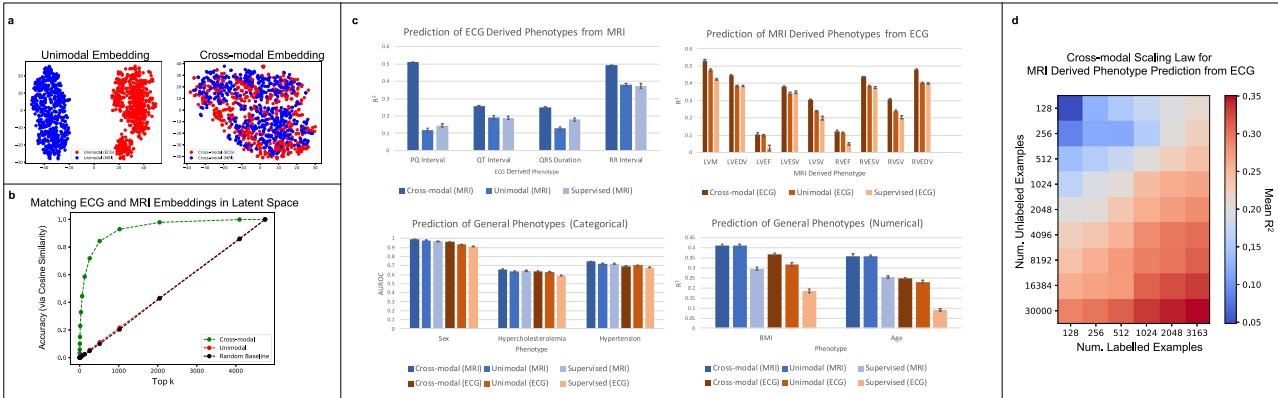

**Fig. 2 | Improvement of phenotype prediction from cross-modal representations over unimodal representations or supervised learning from the original modalities. a** A t-SNE visualization of the cross-modal embeddings for the ECG and MRI samples demonstrates that the modality specifc embeddings are well-mixed, unlike the modality specific embeddings obtained from the unimodal auto-encoders. **b** Ranking each MRI by its cosine similarity with a given ECG in the latent space, we visualize the accuracy that the ground truth MRI appears in the top $k$ neighbors among 4752 test ECG-MRI pairs from the UK Biobank. **c** Kernel regression on cross-modal representations outperforms kernel regression on unimodal representations and supervised deep learning methods on 4 different tasks: (1) prediction of ECG derived phenotypes from MRIs only ($n = 4120$, mean values are reported with error bars indicating one standard deviation); (2) prediction of MRI-derived phenotypes from ECG only ($n = 4218$, mean values are reported with error bars indicating one standard deviation); (3) prediction of general physiological

phenotypes that are of categorical nature from either ECG or MRI ($n = 4218$, mean values are reported with error bars indicating one standard deviation); and (4) prediction of general physiological phenotypes that are of continuous nature from either ECG or MRI ($n = 4212$, mean values are reported with error bars indicating one standard deviation). All MRI phenotype abbreviations are defined in the "Methods" subsection "Models, data, and scaling law for phenotype prediction tasks". Error bars are computed using 5-fold cross-validation. **d** Analysis of the scaling law when utilizing our framework for predicting MRI derived phenotypes from ECGs only. We observe that increasing the number of unlabelled ECG−MRI pairs for pre-training boosts the mean $R^2$ prediction of 9 MRI-derived phenotypes by twice as much as increasing the number of labelled MRI samples. This analysis highlights the benefit of collecting more unlabelled ECG−MRI pairs as compared to paired labelled examples for this task.

modal autoencoder trained on paired cardiac MRI and ECG samples, we show that utilizing standard regression methods (e.g. kernel, linear, or logistic regression) for supervised learning on our cross-modal representations leads to improved prediction of (1) MRI-derived phenotypes (e.g. left ventricular mass, right ventricular end-diastolic volume, etc.) from ECG only; (2) ECG derived phenotypes (e.g. length of the PR interval, QT interval, etc.), from MRI only; and (3) prediction of general phenotypes (e.g. age, sex, body mass index, etc.) from either ECG or MRI. We observe that predictive models applied to our cross-modal representations generally outperform supervised deep learning models and supervised learning on traditional unimodal autoencoder representations. In Supplementary Fig. S3, we additionally demonstrate that cross-modal representations outperform *semi-supervised* unimodal autoencoders, i.e., those trained to simultaneously auto-encode and predict labels from the latent space. Overall, our cross-modal embeddings improve the representational power of inexpensive and prevalent ECGs for predicting clinical phenotypes by leveraging just a few MRI samples.

Cross-modal embeddings allow for matching cardiac MRI and ECG test samples. We begin by verifying that our training methodology provides a latent space in which corresponding ECG and MRI pairs are nearby. Hence, even in the absence of one of the modalities, the cross-modal autoencoder provides a representation that is characteristic of all available modalities. In Fig. 2a, we provide a t-distributed stochastic neighbor embedding (t-SNE) visualization comparing the unimodal and cross-modal autoencoder latent space representations for 500 paired ECG and MRI test samples. We can use a combined t-SNE visualization of the two modalities also for the latent space embedding obtained from the unimodal autoencoders, since our ECG and MRI autoencoders both use latent embeddings of the same size (256 dimensions). The t-SNE plots demonstrate that the ECG and MRI samples are well-mixed in the cross-modal latent space, while the two are clearly separated in the corresponding unimodal latent space. To quantify the benefit of cross-modal representations, we compute the accuracy that the correct MRI pair lies within the top $k$ nearest

neighbors (under cosine similarity) for 4752 test ECGs across embeddings from cross-modal autoencoders, unimodal autoencoders, and a baseline where ECGs and MRIs are randomly paired. Figure 2b demonstrates that cross-modal representations outperform unimodal representations in this task, with the latter performing similarly to the random baseline.

Cross-modal representations improve phenotype prediction from a single modality. We now show that our learned cross-modal representations are more effective for downstream phenotype prediction than unimodal representations or supervised deep learning methods. In particular, we consider four groups of phenotype prediction tasks: prediction of (1) continuous valued MRI-derived phenotypes from ECG; (2) continuous valued ECG-derived phenotypes from MRI; (3) categorical physiological phenotypes from either ECG or MRI; and (4) continuous physiological phenotypes from either ECG or MRI (see Fig. 2c. All MRI-and ECG-derived phenotypes as well as the categorical and continuous-valued physiological phenotypes were provided by the UK Biobank. For all prediction tasks, we utilized the same training, validation, and held out test data from the UK Biobank. Importantly, we note that all data considered for the downstream prediction tasks were excluded from the training procedure for the cross-modal autoencoders. This is critical since otherwise, we could simply train a cross-modal autoencoder to zero error on paired data, and our learned representations would naturally benefit from using both MRI and ECG features for any downstream prediction task; see the "Methods" subsection "Models, data, and scaling law for phenotype prediction tasks" for details on the data splits considered. Again, we note that only a single modality is used for each of these tasks, i.e., we are not giving the cross-modal autoencoders access to any paired samples for the downstream phenotype prediction tasks.

Our results demonstrate the value of cross-modal embeddings for improving the prediction of clinical phenotypes including diseases such as left ventricular hypertrophy (LVH), left ventricular systolic dysfunction (LVSD), and hypercholesterolemia. We utilize kernel regression to perform supervised learning from the cross-modal and

unimodal embeddings; see Supplementary Fig. S4 for a comparison with the performance of linear regression and logistic regression. For fair comparison with supervised deep learning models, we extract the embeddings given by the last layer of the trained neural networks and apply kernel regression on these embeddings; see the "Methods" subsection "Models, data, and scaling law for phenotype prediction tasks" for a description of the architectures for all deep networks used in this task. In all but one setting (hypertension classification), we observe that predictions from our cross-modal latent space improve over predictions from unimodal latent spaces and those from direct supervised learning methods. An important practical implication of these results is that our method is capable of improving the prediction of a variety of phenotypes just using ECGs, which are far easier to obtain and more plentiful than MRIs. This is exemplified by the improvement in prediction of MRI derived phenotypes from cross-modal embeddings of ECGs shown in Fig. 2c.

To further evaluate the results described above, in the "Methods" subsection "Models, data, and scaling law for phenotype prediction tasks" and Supplementary Fig. S5, we additionally consider the prediction of cardiovascular diseases based on thresholded MRI-derived phenotypes. In addition, in Supplementary Fig. S6, we showcase the impact of incorporating circulating biomarkers such as C-reactive protein (CRP) and low-density lipoproteins (LDL) on phenotype prediction. In general, we find that CRP and LDL improve performance for predicting age and BMI. Furthermore, in Supplementary Fig. S7, we demonstrate that there is a boost in the prediction of MRI-derived phenotypes when stratifying phenotypes by sex and BMI. Lastly, in Supplementary Fig. S8, we demonstrate that the cross-modal representation can improve prediction for diseases such as atrial fibrillation (AF) or heart failure (HF) using labels provided by the UK Biobank.

Increasing the number of unlabelled samples improves the prediction of MRI-derived phenotypes from cross-modal ECG representations. We now analyze the relationship between the amount of labelled data for supervised learning, the amount of unlabelled data for cross-modal autoencoding, and the performance of supervised learning from cross-modal latent representations. Such an analysis is crucial for understanding the number of labelled and unlabelled data samples needed to build an effective cross-modal autoencoder for use in practice. In Fig. 2d, we focus on such an analysis for the practically relevant setting of predicting MRI derived phenotypes from ECGs. In particular, we measure the mean $R^2$ performance across all 9 MRI derived phenotypes from Fig. 2c as a function of the number of unlabelled samples for autoencoding and labelled samples for supervised learning from cross-modal embeddings. Performing a scaling law analysis (see the "Methods" subsection "Cross-modal autoencoder architecture and training details"), we observe that collecting unlabelled samples for autoencoding leads to roughly twice the increase in predictive performance as collecting labelled samples for supervised learning. Since the collection of unlabelled ECG–MRI pairs is easier than the collection of labelled MRIs, our cross-modal autoencoder is able to leverage easily collectable data to improve the performance on these downstream phenotype prediction tasks.

### Cross-modal autoencoder framework enables generating cardiac MRIs from ECGs

Our framework enables the translation of ECGs, an easy-to-acquire modality, to cardiac MRIs, a more expensive, difficult-to-acquire modality. To perform such translation, we simply provide test ECGs into our ECG-specific encoder and then apply the MRI-specific decoder to translate from ECGs to MRIs. We note that since the two data modalities capture complementary cardiac features (ECGs capturing myoelectric information and MRIs capturing structural information), such translation is a nontrivial task. Nevertheless, we show that the translation of ECGs provided by a cross-modal autoencoder

remarkably captures features present in MRIs, and we quantify the amount of such features captured via the translation.

Cardiac MRIs generated from test ECGs capture MRI specific phenotypes. We begin by qualitatively analyzing the reconstructions and translations of 12-lead ECG and 50 frame cardiac MRI test pairs using our cross-modal autoencoder. In Fig. 3, we demonstrate that translations from ECGs to MRIs generally capture MRI-derived phenotypes such as left ventricular mass (LVM) or right ventricular end-diastolic volume (RVEDV). In Fig. 3a and b, we consider translating from ECGs to MRIs for test samples of individuals with high or low LVM/RVEDV. We observe that the corresponding translations generally capture whether an individual has high or low LVM/RVEDV, as indicated by the annotated regions in red. For comparison, we additionally present reconstructions given by our model when provided the test MRI as an input. These reconstructions demonstrate that the MRI-specific decoder has the capacity to reconstruct fine grained details of an MRI. Hence, the difference in quality between reconstructions and translations can be attributed to the difference in embedding provided from ECG and MRI specific encoders. Additional translations from ECG to MRI (and vice-versa) are presented in Supplementary Figs. S9 and S10, demonstrating that decoding ECG or MRI cross-modal embeddings after shifting them in a direction of phenotypic effect (e.g. moving from low LVM to high LVM) leads to the desired phenotypic effect on the original modality (e.g. increased LVM in the corresponding generated MRI). Videos of reconstructed MRIs, translated ECGs, and MRIs shifted according to a selected phenotypic effect are presented in Supplementary Videos S1–S5.

In order to quantify the effectiveness of the translations using the cross-modal autoencoder, we compare the predictions of the translation to a neural network directly trained to predict LVM and RVEDV on the original modality. In particular, in Fig. 3c, we verify that the prediction of LVM and RVEDV from the reconstructed and translated MRIs positively correlates with that from ground truth MRIs. Hence, the translations of test ECGs provided by our cross-modal autoencoder indeed generally capture MRI derived phenotypes, as shown in Fig. 3a and b.

### Cross-modal autoencoder framework enables genome-wide association study using integrated latent space

Next, we analyze whether cross-modal embeddings can be used to identify genotype–phenotype associations related to the heart. As a first step, we verify that performing a GWAS on labels derived from cross-modal representations leads to the recovery of SNPs previously associated with common disease phenotypes. We then develop a method based on the cross-model latent space to perform GWAS in the absence of labelled phenotypes, i.e. an unsupervised GWAS. We demonstrate that our unsupervised GWAS approach applied to cross-modal embeddings recovers SNPs typically identified by performing GWAS on labelled data, as well as those found in more computationally demanding ECG-wide screens[40].

GWAS of phenotypes predicted from cross-modal representations recovers phenotype-specific SNPs. In order to verify that cross-modal representations capture genetic associations with respect to a specific phenotype, we perform a GWAS on single trait predictions based on these representations; see the "Methods" subsection "GWAS of phenotypes derived from cross-modal representations" for a description of performing such GWAS and a list of confounders considered. As an example, the Manhattan plot in Fig. 4a shows that such GWAS for body mass index (BMI) predicted from cross-modal embeddings identifies the gene FTO, which is known to have an effect on BMI and obesity risk[41,42]. Similarly, performing a GWAS of right ventricular ejection fraction (RVEF) predicted from MRI cross-modal representations identifies lead SNPs corresponding to genes BAG3, HMGA2, and MLF1, which have all been previously associated

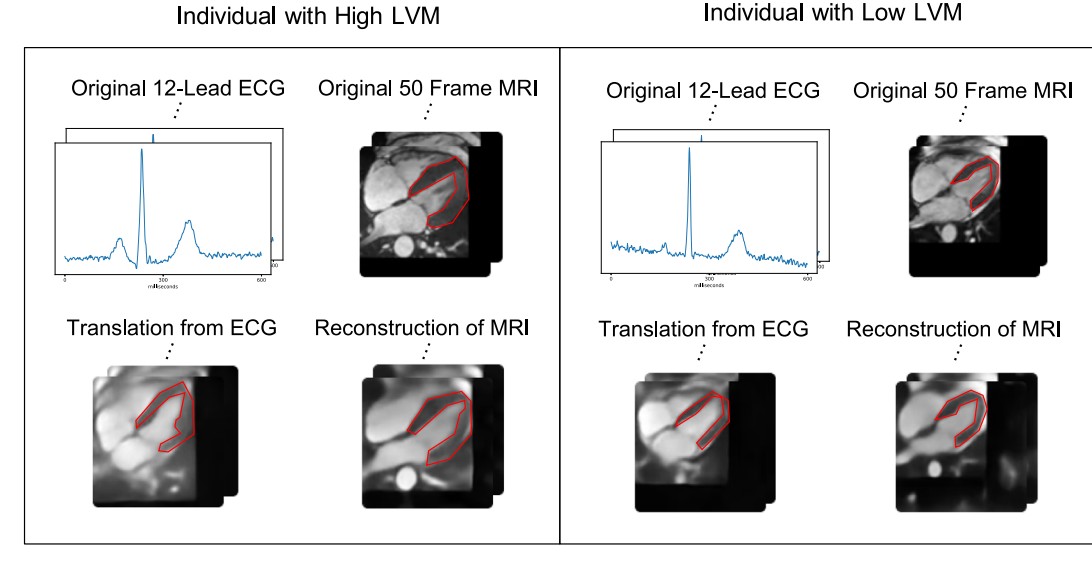

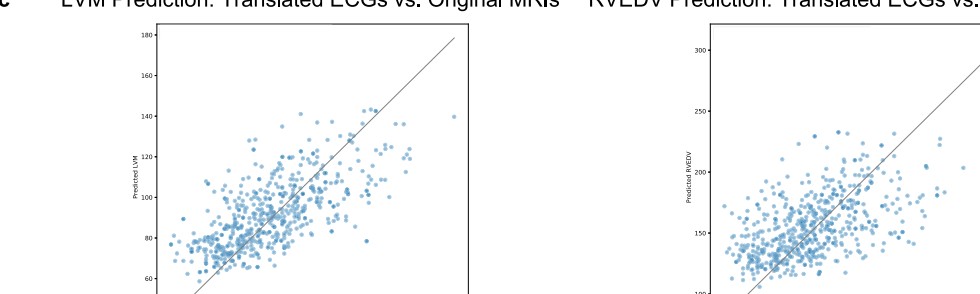

**Fig. 3 | Cross-modal autoencoders enable imputing cardiac MRIs from ECGs while capturing MRI-specific features such as left ventricular mass (LVM) and right ventricular end-diastolic volume (RVEDV) on test MRI–ECG pairs.**
**a** Examples showing qualitatively that MRIs imputed from test ECG samples capture LVM for those individuals with LVM in the highest and lowest quartile. The LVM in the original, translated, and reconstructed MRI is shown in red. **b** Examples showing qualitatively that MRIs imputed from test ECGs capture RVEDV for those individuals with RVEDV in the highest and lowest quartile. The RVEDV in the original, translated, and reconstructed MRI is shown in red. **c** The predictions of LVM and RVEDV on MRIs imputed from test ECGs correlate with the predictions of these phenotypes performed on the original MRIs.

with RVEF[43]. These results indicate that our learned representations are physiologically meaningful. Additional examples of ECG phenotypes derived from cross-modal embeddings are presented in Supplementary Fig. S11.

Unsupervised GWAS of cross-modal representations leads to the recovery of SNPs associated with a given modality. While we have so far demonstrated that we can perform GWAS on one-dimensional traits using our cross-modal embedding, we note two limitations of this

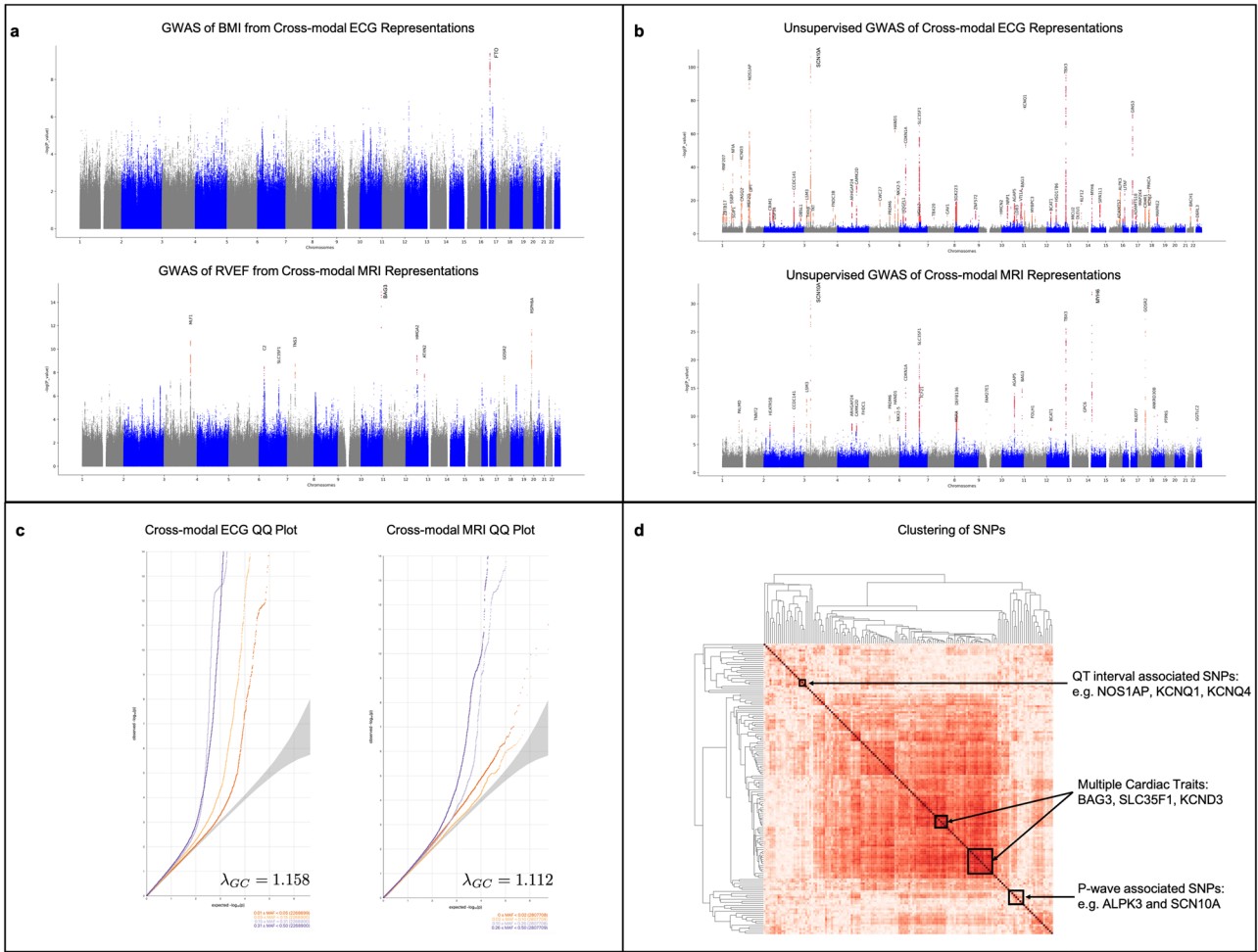

**Fig. 4 | Cross-modal autoencoders capture genotype–phenotype associations for cardiovascular data. a** Manhattan plots for GWAS of BMI and RVEF derived from cross-modal embeddings identify lead SNPs associated with these traits. For BMI, such GWAS identifies SNPs associated with FTO, which is known to have an effect on BMI and obesity risk. For RVEF, such GWAS identifies SNPs associated with BAG3, HMGA2, and MLF1, which have been previously associated with RVEF. **b** To more generally capture genetic associations with the heart, a GWAS can be performed in the cross-modal ECG and MRI latent space even in the absence of labelled data. The Manhattan plots of such unsupervised GWAS identify lead SNPs including those associated with NOS1AP, TTN, SCN10A, SLC35F1, KCNQ1, which have been previously associated with cardiovascular phenotypes. **c** The corresponding QQ plots and $\lambda_{GC}$ factors indicate that there is minimal inflation in the unsupervised GWAS of cross-modal ECG and cardiac MRI embeddings. **d** Clustering SNPs by the vector from the mean embedding of homozygous reference samples to the mean embedding of heterozygous and homozygous alternate samples in order to group SNPs by similar phenotypic effect results in clusters of SNPs corresponding to those associated with the QT interval (NOS1AP and KCNQ1), those related to the P-wave (SCN10A and ALPK3), as well as SNPs that affect multiple cardiac traits (e.g., BAG3, SLC35F1, and KCND3).

approach. The first limitation is that genetic variants are often pleiotropic, affecting many traits at the same time. Indeed, a variant can have a small effect on a pair of traits, and looking at one trait alone would not provide sufficient power for identifying this variant. The second limitation is that one-dimensional traits, even in aggregate, are an incomplete characterization of the information in a diagnostic modality. For example, in ECGs, we aim to not only identify variants that impact the measured phenotypes such as the QT interval but variants that affect the ECG in any way. In the following, we, therefore, develop an unsupervised GWAS methodology that provides a principled approach for automatically discovering genetic variants from rich diagnostic modalities without using any labelled phenotype measurements or turning to manual, time-intensive interpretability strategies such as saliency maps or direct visualization of ECGs and MRIs. Our approach is as follows: (1) for each SNP, we identify those individuals in the latent space that are either homozygous reference, heterozygous, or homozygous alternate; (2) we then ask whether these distributions are separable, and quantify the level of separation among these groups via a *p*-value from a Multivariate Analysis of Variance

(MANOVA). The statistics used for computing p-values using MANOVA are discussed in the "Methods" subsection "GWAS of phenotypes derived from cross-modal representations". Importantly, prior to performing MANOVA across the sets, we need to account for potential genetic confounders (stratification), which are inherently reflected in the cross-modal embeddings. Without accounting for confounders, MANOVA is unable to recover the genetic signals from the latent embeddings (see Supplementary Fig. S13a, b). To remove the effect of possible confounders, we use the iterated nullspace projection (INLP) method[38]. This method was originally developed in the natural language processing domain to protect against biases (such as gender stereotypes) from appearing in word embeddings. We leverage INLP in the medical domain to ensure that confounders such as principal components of ancestry cannot be easily predicted from cross-modal embeddings, thereby ensuring that such features do not arise as confounders in the GWAS. This method iteratively removes dimensions from the latent space until the remaining embeddings cannot be used to predict any confounder (see the "Methods" subsection "Unsupervised GWAS of cross-modal representations" for additional

details). After removing latent space dimensions that are predictive of confounders using INLP, we utilize MANOVA on the lower dimensional embeddings to perform an unsupervised GWAS (see the "Methods" subsection "Unsupervised GWAS of cross-modal representations" for a list of confounders considered).

In Fig. 4b, we visualize the Manhattan plots resulting from utilizing our unsupervised GWAS approach on the cross-modal ECG and MRI embeddings. We observe that lead SNPs such as NOS1AP[44,45], TTN[46,47], SCN10A[48–50], SLC35F1[51], and KCNQ1[52,53] are consistent with those identified by prior work. In Supplementary Fig. S12, we present the results of the unsupervised GWAS performed on the joint cross-modal embeddings of both ECG and MRI along with unsupervised GWAS performed on the unimodal autoencoder representations for these modalities. Full lists of the lead SNPs identified in each analysis are presented in Supplementary Tables S1–S3. Furthermore, Fig. 4c shows the QQ plots and the corresponding $\lambda_{GC}$ values to verify that the corresponding p-values are not inflated after removing the effect of confounders via INLP. In Supplementary Fig. S13, we analyze the impact of varying hyper-parameters of INLP on the level of inflation present in the resulting GWAS. As shown in Supplementary Fig. S14, the lead SNPs identified from our GWAS approach for cross-modal ECGs generally include those from GWAS on individual ECG phenotypes. In Supplementary Fig. S15, we provide Venn diagrams comparing the SNPs found by the unsupervised GWAS and those found by the supervised GWAS on ECG-derived phenotypes. We also identify many sites not previously associated with ECG or MRI traits, but which have clear associations with the cardiovascular system in general, for example NRP1, previously associated with HDL cholesterol[54], USP34 previously associated with cardiovascular disease[55], and NRG1 previously associated with systolic blood pressure[56]. We note that the cross-modal MRI GWAS identifies fewer lead SNPs than the cross-modal ECG GWAS, which could be because MRIs are more strongly associated with the confounders and thus removing confounders via INLP may also remove genetic signal. Indeed, confounders such as age and sex are much more easily predicted from cross-modal MRI embeddings than cross-modal ECG, as is showcased in Fig. 2c. To illustrate the difference between between unsupervised GWAS of different representations, we compare the corresponding differences in Manhattan plots in Supplementary Fig. S16.

Clustering SNPs in the cross-modal latent space identifies SNPs with similar phenotypic impact. An additional benefit of our cross-modal approach for genetic discovery is that we can cluster SNPs in the latent space to group those with similar phenotypic effects. In particular, we perform hierarchical clustering based on the direction from the mean embedding of the homozygous reference group to the mean embedding of the heterozygous and homozygous alternate groups for any given SNP (see Fig. 1d). In Fig. 4d, we analyze the SNP clusters given by performing hierarchical clustering on the SNP signatures in the cross-modal embeddings given only ECG inputs (see the "Methods" subsection "GWAS of phenotypes derived from cross-modal representations" for details regarding hierarchical clustering). In particular, we find two clusters corresponding to SNPs affecting the QT interval (SNPs associated with NOS1AP and KCNQ1) and SNPs related to the P-wave (SNPs associated with SCN10A and ALPK3) of the ECG. We find several additional clusters corresponding to SNPs affecting multiple cardiac traits such as those associated with BAG3, SLC35F1, or KCND3. Supplementary Fig. S17 shows a high resolution version of this clustering and a clustering of a subset of lead SNPs, which illustrates robustness of our clusters.

## Discussion

In this work, we developed a cross-modal autoencoder framework for integrating data across multiple modalities to learn holistic representations of the physiological state. Using the heart as a model system, we integrated cardiac MRI and ECG data to showcase the benefit of cross-modal representations via the following three applications: (1) improving prediction of phenotypes from a single modality; (2) enabling imputation of hard-to-acquire modalities like MRIs from easy-to-acquire ECGs; and (3) identifying genotype associations with general cardiovascular phenotypes. In particular, we showed that cross-modal representations improve the prediction of cardiovascular phenotypes from ECGs alone. This setting is of practical importance given the abundance of ECG data over more difficult-to-acquire modalities such as MRI. Interestingly, we observed that increasing the number of unlabelled ECG and MRI pairs was more beneficial than increasing the number of labelled MRI data. We also demonstrated that cross-modal autoencoders enable imputing cardiac MRIs from ECGs. Importantly, we showed that the MRI-derived phenotypes are conserved in the translation. We also showed that the cross-modal representations can be used to perform GWAS. Notably, such an analysis not only recovers known phenotype-specific SNPs but can also be used to perform unsupervised GWAS to identify SNPs that generally affect the cardiovascular system. The proposed unsupervised GWAS method provides an effective and efficient approach to genetic discovery as it has the same computational cost as performing a single GWAS and, in contrast to existing methods for GWAS, it does not require any labelled phenotype data.

The reliable performance boost in phenotype prediction from a cross-modal embedding further highlights its applicability to aid in diagnostics. An interesting future application of our framework is to determine the extent to which cross-modal ECG embeddings can be translated to a hospital setting and, in conjunction with other biomarkers, improve the prediction of specific cardiovascular diseases. However, we acknowledge that a current limitation of our work is that UK Biobank samples are limited in their diversity with individuals primarily falling between the ages of 40–69. In addition, the UK Biobank is known to contain racial and socioeconomic biases, which can lead to problematic inequities in terms of healthcare[57]. It would therefore be important to re-train or update our model on a more diverse population and perform a careful analysis of how well our model generalizes to underrepresented cohorts in future work before translating this method to hospital settings. For deployment in such settings, it is critical to account for potential confounding factors. For example, in our dataset, LDL was negatively correlated with the incidence of hypercholesterolemia, which is presumably due to these individuals taking lipid-lowering medications. While we showed how our cross-modal embedding of ECG and cardiac MRI data can be used to improve the prediction of clinical phenotypes such as LVH, LVSD, and hypercholesterolemia, our framework is general and only requires a few labelled samples to be applicable to other clinical phenotypes. As such, an interesting direction for future research is to understand the extent to which related neurological phenotypes can be predicted from cross-modal ECG and MRI embeddings.

Our unsupervised GWAS was able to leverage information across ECGs and cardiac MRIs to capture a wide range of SNPs that had an impact on the cardiovascular system. Subsets of these SNPs were previously found by traditional supervised approaches on individual phenotypes. Given that our approach is cross-modal, we also identified SNPs that had not been found by previous GWAS approaches. This framework for performing unsupervised GWAS in cross-modal representations also opens important avenues for future work. Investigating the differences in the identified SNPs between unsupervised and traditional GWAS is an interesting direction for future work and requires careful consideration of potential confounders. In particular, since cross-modal autoencoders learn representations from modalities directly, confounders are typically embedded in the representations. Indeed, we for example observed that MRI cross-modal embeddings can predict sex and age effectively. To minimize the effect of such confounders when performing genetic analyses, we were stringent in adjusting the latent space such that one could no longer predict

confounders effectively from the learned representations. Developing more causally-grounded methods for confounder removal from a cross-modal latent space is an important open problem. Moreover, via a simple clustering of cross-modal embeddings, our framework allows for grouping SNPs by phenotypic effect without the need of labelled phenotypes. Since our framework can be used to integrate any number of data modalities, an exciting direction of future work is to use such modalities in other organs to better characterize the effect of SNPs with similar signatures in an unsupervised manner. Such identification requires reliable translation from the cross-modal latent space into different modalities. While we showed that our framework is capable of translating from easy-to-collect ECGs to more difficult-to-collect MRIs while preserving relevant features, an interesting direction of future work is to understand how far such translations can be pushed.

With the rise of Biobanks around the world, our cross-modal integration framework opens an important avenue to integrate multiple modalities to build better representations of patient physiological state and thereby have an important impact on diagnostics and genomics. While we demonstrated the effectiveness of our cross-modal autoencoder framework on the cardiovascular system, our framework is broadly applicable to other organ systems.

## Methods
Our research complies with all relevant ethical regulations. Access was provided under UK Biobank application #7089. Analysis was approved by the Broad Institute institutional review board.

### Study design
All analyses were performed on the UK Biobank, a prospective cohort of over 500,000 healthy adults that were aged 40–69 during enrollment, which took place from 2006 to 2010. At the time of our analysis the UK Biobank had released cardiovascular magnetic resonance imaging for over 44,644 participants, 38,686 of whom also had a 12-lead 10-second resting ECG acquired on the same day. The ECG and MRI data for an individual are collected in the same assessment center. While different MRI views were obtained, we only considered the 4-chamber long axis view with balanced steady-state free-precession cines, containing 50 frames throughout the cardiac cycle. The ECG data also spanned a single cardiac cycle, as we used the 1.2-s median waveforms (600 voltages) derived from the full 10-s ECG. All voltages were transformed to millivolts, and all MRI pixels were normalized to have mean 0 and standard deviation 1 for each individual. The MRIs were cropped to the smallest bounding box which contained all cardiac tissues in all 50 frames as determined by the semantic segmentation in[58]. Labels for the sex of individuals were provided by UK Biobank. In particular, of the samples used for training and evaluating autoencoders that had cardiac MRI available, 21,066 individuals were labelled by UK Biobank as genetic sex of male and 23,577 individuals were labelled as genetic sex of female.

### Cross-modal autoencoder architecture and training details
**Model architecture.** The modality-specific encoders and decoders used in this work were selected through Bayesian hyper-parameter optimization[59]. In particular, we used a base architecture of densely connected parallel convolutional blocks[60,61] with 1d convolutional layers for ECGs and 2d convolutional layers for MRIs. For modality-specific modals, we optimized over the width, depth, activation functions, regularization and normalization strategies to achieve minimum reconstruction error for a given maximum overall capacity of 10 million parameters and a 256 dimensional latent space. Since optimization occurs for each modality independently, encoding, decoding, and pairing are distinct tasks and can be trained asynchronously and distributed across machines. We note that simpler architectures such as those from ref. 62 are also usable in our framework, but we observed that the optimized models showed improvements in convergence

speed, reconstruction, and latent space utility for downstream tasks (see Supplementary Fig. S1).

To ensure that only one modality is needed at test time, we additionally utilized dropout[63] to merge modality-specific embeddings. In particular, during training, we employed dropout of a random subset of coordinates of the ECG embedding and merged it with the complementary coordinates from the MRI embedding. The resulting merged embedding was then decoded to reconstruct the original ECG and MRI examples. We note that other techniques such as averaging or concatenation to merge modality-specific embeddings were less effective than dropout, and other losses to pair modalities such as maximizing cosine similarity or minimizing Euclidean distance between paired samples were less effective than using a contrastive loss (see Supplementary Fig. S1).

**Training methodology.** Let $X^{(j)} = \{x^{(i,j)}\}_{i=1}^{n} \subset \mathbb{R}^{d_j}$ denote the set of samples of modality $j$ for $j \in [m]$ where $[m] = \{1, 2, \ldots m\}$, and let $n$ denote the number of samples and $d_j$ the dimension of modality $j$. Consider the paired setting where the samples $\{x^{(i,j)}\}_{j=1}^{m}$ correspond to multiple data modalities for the same sample (e.g. cardiac MRI and ECG for the same individual). Given a subset of these modalities $\{x^{(i,j)}\}_{j \in \mathcal{I}}$ for $\mathcal{I} \subset [m]$, we constructed a cross-modal autoencoder that produces the remaining representations $\{x^{(i,j)}\}_{j \in [m] - \mathcal{I}}$ as follows. We decomposed our model into encoders $\{f_j : \mathbb{R}^{d_j} \to \mathcal{Z}\}_{j=1}^{m}$ and decoders $\{g_j : \mathcal{Z} \to \mathbb{R}^{d_j}\}_{j=1}^{m}$, where the functions $f_j$ and $g_j$ are parameterized using deep neural networks. The neural networks were trained to pair and reconstruct each data modality. Modality-specific encoders and decoders allowed for inferring all modalities given any single one.

The training loss, $\mathcal{L}$, for cross-modal autoencoders is given as the linear combination of the following two losses: (1) a reconstruction loss, $L_{\text{Reconstruct}}$, which is used to reconstruct the original modalities; and (2) a representation loss $L_{\text{Contrast}}$, which is used to ensure that the representations for modalities corresponding to the same sample are embedded nearby in the latent space. We now provide a formal definition of these loss functions:

$$\mathcal{L}(\{X^{(j)}, f_j, g_j\}) = L_{\text{Contrast}}(\{X^{(j)}, f_j\}) + \lambda L_{\text{Reconstruct}}(\{X^{(j)}, f_j, g_j\}), \quad (1)$$

$$L_{\text{Reconstruct}}(\{X^{(j)}, f_j, g_j\}) = \sum_{i=1}^{n} \sum_{j=1}^{m} \| x^{(i,j)} - g_j(f_j(x^{(i,j)})) \|^2, \quad (2)$$

$$
\begin{aligned}
L_{\text{Contrast}}(\{X^{(j)}, f_j\}) = -\frac{1}{2} \sum_{I_k \in P_b} \sum_{\substack{j_1, j_2 = 1 \\ j_1 \neq j_2}}^{m} \sum_{i=1}^{|I_k|} &\log\left( \frac{\exp\left(e^{\text{temp}} f_{j_1}(x^{(i,j_1)}) \cdot f_{j_2}(x^{(i,j_2)})\right)}{\sum_{i'=1}^{|I_k|} \exp\left(e^{\text{temp}} f_{j_1}(x^{(i',j_1)}) \cdot f_{j_2}(x^{(i,j_2)})\right)} \right) \\
&+ \log\left( \frac{\exp\left(e^{\text{temp}} f_{j_1}(x^{(i,j_1)}) \cdot f_{j_2}(x^{(i,j_2)})\right)}{\sum_{i'=1}^{|I_k|} \exp\left(e^{\text{temp}} f_{j_1}(x^{(i,j_1)}) \cdot f_{j_2}(x^{(i',j_2)})\right)} \right),
\end{aligned}
$$

$$(3)$$

where $\lambda$ is a hyperparameter to balance the losses, temp is a trainable temperature scalar as in ref. 64, and given a batch size $b$, $P_b = \{I_1, \ldots, I_{\lceil \frac{m}{b} \rceil}\}$ denotes a partition of $[m]$ such that $|I_\ell| = b$ for $\ell \in [\lfloor \frac{m}{b} \rfloor]$. Intuitively, the contrastive loss above pushes embeddings from the same individual and different modalities closer together while pulling apart embeddings of different individuals and different modalities.

In our experiments, we used a batch size of 4 samples ($b = 4$) and used $\lambda = 0.1$. All models were optimized with the Adam optimizer[65] and a learning rate of 1e−3 for unimodal autoencoder training and 2e−5 for cross-modal fine-tuning. The learning rate was decayed by a factor of 2 after each epoch without an improvement in validation loss and after 3

decays optimization was terminated. Samples were split into a training set of 27,160 individuals, a validation set of 6780, and a test set of 4746. To avoid any data leakage the entire test set was made of samples with MRI phenotypes labelled in[39], ensuring a fair comparison with models trained on downstream prediction tasks. The MRIs were normalized per individual to have mean zero and standard deviation 1. All ECG readings were converted to millivolts before training.

**Software for training cross-modal autoencoders.** We used numpy (version 1.22.4)[66] and tensorflow (version 2.9.1)[67] for training cross-modal autoencoders.

**Models, data, and scaling law for phenotype prediction tasks**
**Supervised learning models for phenotype prediction tasks.** We compared phenotype prediction from cross-modal embeddings to training supervised models with the same encoder architecture as described in the section "Cross-modal autoencoder architecture and training details". In particular, we trained these supervised models for phenotype prediction by adding a last layer and updating the weights via a logcosh loss for continuous tasks and cross entropy loss for categorical tasks. We also used the same optimization procedures for the hyper-parameters and the same stopping criteria as described in the section "Cross-modal autoencoder architecture and training details".

**Definitions of MRI phenotype abbreviations considered in downstream prediction tasks.** The following standard abbreviations were used for MRI-derived phenotypes. Left ventricular mass was denoted LVM, left ventricular end diastolic volume was denoted LVEDV, left ventricular ejection fraction was denoted LVEF, left ventricular end systolic volume was denoted LVESV, left ventricular stroke volume was denoted LVSV, right ventricular ejection fraction was denoted RVEF, right ventricular end systolic volume was denoted RVEDV, and right ventricular end diastolic volume was denoted RVEDV.

**Data splits for phenotype prediction tasks.** For all phenotype prediction tasks, we only considered data that was held out during cross-modal autoencoder training. This is crucial since otherwise the autoencoder would automatically utilize both MRI and ECG data for all phenotype predictions and thus naturally perform better than prediction from any individual modality. Since we were limited by the availability of labelled data for MRI derived phenotypes, we held out all data for which there was an available MRI derived phenotype from the autoencoder training and validation set. In particular, even though we may have phenotypes such as age or sex for all individuals, we only measured performance on phenotypes for the held out test set to ensure fair comparison with other models. Naturally, using all available labelled samples for predicting sex or age would have boosted performance, but would have given an unfair advantage to our method. This left us with 4218 samples containing MRI derived phenotypes. For MRI derived phenotype prediction, we split these into 3163 samples for training, 527 for validation, and 528 for test. Only 4120 of these samples had corresponding ECG derived phenotypes, and so we used 3083 of these for training, 516 for validation, and 521 for test. For categorical general phenotypes, we used the same splits as those for MRI-derived phenotypes. For continuous valued general phenotypes, we considered only the subset of the 4218 samples that had labels available. In particular, we used 3158 samples for training, 527 for validation, and 527 for testing.

**Linear, logistic, and kernel regression models for phenotype prediction tasks.** For phenotype prediction from latent space embeddings, we considered the performance of three models (1) kernel regression with the Neural Tangent Kernel (NTK)[68]; (2) linear regression ; and (3) logistic regression. We considered the NTK since it was shown to have superior performance on supervised learning problems[69,70]. For the prediction of MRI derived phenotypes, ECG derived phenotypes, or continuous general phenotypes, we measured performance using $R^2$, and we compared the performance of the NTK and linear regression. For the prediction of categorical phenotypes, we measured performance using the area under the receiver operator characteristic curve (AUROC), and we compared the performance of the NTK and logistic regression. We utilized EigenPro[71] to solve kernel ridge-less regression and linear regression. We used the validation splits to select the early-stopping point for the EigenPro iteration. Similar results can be obtained with $\ell_2$-regularized kernel and linear regression using the Scikit-learn implementation[72], but require the more computationally demanding step of fine-tuning of the regularization parameter based on validation performance. For classification tasks, we used the implementation of $\ell_2$-regularized logistic regression from ref. [72], and we applied the following weighting on the loss to account for class imbalances: if there were $n$ total samples of which $r$ had label 1, then we weighted the loss for these samples by $\frac{n}{r}$ and the loss for the samples with label 0 by $\frac{n}{n-r}$.

**Scaling law for prediction of MRI-derived phenotypes from cross-modal ECG representations.** We now describe our scaling law analysis used to determine the relationship between the amount of labelled data for supervised learning (denoted by $v$), the amount of unlabelled data for cross-modal autoencoding (denoted by $u$), and the performance of supervised learning from cross-modal latent representations (denoted by $r$). We used linear regression to map from $(\log_2 v, \log_2 u)$ to $r$ for the 54 samples considered in Fig. 2d. The corresponding linear mapping is given by:

$$r = 0.0158\log_2 u + 0.007\log_2 v,$$

and yields $R^2 = 0.983$. Hence for these tasks, we were able to reliably predict the boost in performance from supervised models on cross-modal embeddings when varying the number of unlabelled ECG–MRI pairs and labelled MRIs. Note that the coefficient of $\log_2 u$ is over twice that of $\log_2 v$ implying that collecting unlabelled ECG–MRI pairs leads to roughly twice the increase in predictive performance as collecting labelled samples.

**Prediction of left ventricular hypertrophy and left ventricular systolic dysfunction.** We used LVM to derive thresholds for left ventricular hypertrophy (LVH) and LVEF to derive thresholds for left ventricular systolic dysfunction (LVSD). To provide a binarized label for LVH, we first normalized all LVM measurements by dividing by body surface area (derived using the Mosteller method). We then stratified by sex and set the LVH label to be 1 if the normalized LVM was greater than 72 if the sex was male, respectively 1 if the normalized LVM was greater than 55 if the sex was female[73]. Supplementary Fig. S5 shows that using logistic regression from cross-modal embeddings leads to the highest AUROC of 0.756 for predicting LVH. For LVSD, the binarized label was obtained as an indicator of whether the LVEF was less than 45%. Again, logistic regression from cross-modal embeddings leads to the highest AUROC of 0.572. In both analyses, standard deviations were computed over 10-fold cross-validation.

**Software for training predictive models.** We used numpy (version 1.21.2)[66], sklearn (version 0.24.2)[72], scipy (version 1.7.1)[74], and pytorch (version 1.9.11) with gpu support (cudatoolkit version 11.1.74)[75] for training predictive models. We used pandas (version 1.3.3)[76,77] for loading data and matplotlib (version 3.4.2)[78] for generating plots.

**GWAS of phenotypes derived from cross-modal representations**
**GWAS on phenotypes predicted from latent representations.** We found multi-pathway genetic signals in the cross-modal latent spaces by analyzing the inferences of the kernel regression models described

above. Specifically, we trained ridge-regression models to use modality-specific cross-modal embeddings to predict ECG phenotypes (e.g. PR Interval $N = 36,645$), MRI-derived phenotypes (e.g. RVEF $N = 4788$) and general demographics (e.g. BMI $N = 38,000$). These simple models endow these GWAS with much greater statistical power, since phenotypes can be predicted for the whole cohort, not just those with labels, as described in ref. 43. For example, GWAS of the less than 5000 MRI phenotypes returned by Petersen et al.[39] yield no genome wide significant hits, while inferences from ridge-regression yield dozens of plausible sites. These sites are confirmed by GWAS of traits computed from semantic segmentation described in ref. 43. Models were fit with 80% of the available labels and evaluated on the remaining 20% and then inferred on the entire cohort.

**Confounders considered in GWAS.** To account for population structure and ascertainment biases all GWAS were adjusted for the top 20 principal components of ancestry, the UK Biobank assessment center where the measurements were conducted, the genomic array batch, as well as age and sex of each individual.

### Unsupervised GWAS of cross-modal representations

**Application of iterative nullspace projection for removing confounders.** To remove the effect of confounders, we utilized the idea of iterative nullspace projection from Ravfogel et al.[79]. Intuitively, this algorithm reduces the dimensionality of the latent space by removing dimensions that are useful for the prediction of confounders. Unlike the original implementation, which is designed primarily for categorical confounder removal and has additional memory overhead from storing projection matrices, we here present an implementation for continuous confounder removal that avoids extra overhead by utilizing the singular value decomposition (SVD). At a high-level, the algorithm involves iterating the following steps until the $R^2$ from step 1 is below a pre-selected threshold (we used $R^2 < 0.001$).

Step 1: Use linear regression to learn a mapping from cross-modal latent embeddings to confounders.
Step 2: Use the singular value decomposition to construct a projection matrix that projects onto the directions of the cross-modal space that are least useful for confounder prediction, i.e. the nullspace of the predictor from step 1.
Step 3: Multiply the cross-modal embeddings by the projection matrix found in step 2.

Mathematically, these steps are implemented as follows. Let $\{x^{(i)}\}_{i=1}^{n} \subset \mathbb{R}^d$ denote the cross-modal latent space embeddings for $n$ individuals and let $\{F^{(i)}\}_{i=1}^{n} \subset \mathbb{R}^m$ denote the set of $m$ confounders for the $n$ individuals. To correct for confounders, we do the following:

Step 1: Learn the regression coefficients $w \in \mathbb{R}^{m \times d}$ by minimizing the loss:

$$\mathcal{L}(w) = \sum_{i=1}^{n} \| wx^{(i)} - F^{(i)} \|_2^2.$$

Step 2: Let $w = U\Sigma V^T$ given by the SVD, where $V^T \in \mathbb{R}^{d \times d}$. To project out the components corresponding to the confounders, select out the bottom $d-m$ rows of $V^T$ into a matrix $\bar{V}^T \in \mathbb{R}^{d-m \times d}$.
Step 3: Replace each $x^{(i)}$ with $\tilde{x}^{(i)} = \bar{V}^T x^{(i)} \in \mathbb{R}^{d-m}$.

Repeat the above steps until the $R^2$ of the predictor $w$ is lower than a fixed threshold.

**MANOVA $p$-value computation.** The $p$-values reported for unsupervised GWAS are from Pillai's trace test statistic from the MANOVA

computation. The Python statsmodels[80] package was used to perform MANOVA.

**Clustering of SNPs by effect.** Agglomerative clustering with Ward's method, which minimizes the total within-cluster variance, was applied to the matrix of SNP vectors. The python sklearn[72] clustering package was used to derive the clusters and dendrograms.

### Reporting summary

Further information on research design is available in the Nature Portfolio Reporting Summary linked to this article.

## Data availability

All data used in this study were obtained from UK Biobank. UK Biobank data is available to researchers from research institutions following IRB and UK Biobank application approval. UK Biobank data are available to qualified investigators via application at https://www.ukbiobank.ac.uk. GWAS results can be found in the GWAS catalog https://www.ebi.ac.uk/gwas/studies/, and catalog accession ids are GCST90250896 (for cross-modal ECG and cardiac MRI unsupervised GWAS), GCST90250897 (for cross-modal ECG unsupervised GWAS), and GCST90250896 (for cross-modal cardiac MRI unsupervised GWAS).

## Code availability

Serialized encoders, decoders and full autoencoder models are available in the github repository https://github.com/broadinstitute/ml4h/tree/master/model_zoo/dropfuse through[81].

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

## Acknowledgements

A.R. and C.U. were partially supported by NCCIH/NIH (1DP2AT012345), NSF (DMS-1651995), ONR (N00014-22-1-2116), the MIT-IBM Watson AI Lab, AstraZeneca, the Eric and Wendy Schmidt Center at the Broad Institute, and a Simons Investigator Award (to C.U.). S.A.L. was supported by NIH grants (R01HL139731, R01HL157635) and the American Heart Association (18SFRN34250007).

## Author contributions

A.R., S.F.F., A.A.P., S.A.L., and C.U. designed the research and wrote the paper. A.R., S.F.F developed and implemented the algorithms and performed model and data analysis. A.R., S.F.F., S.K., K.N., P.B., S.A.L., A.A.P., and C.U. provided feedback on the model, data analysis, and paper.

## Competing interests

S.A.L. receives sponsored research support from Bristol Myers Squibb, Pfizer, Boehringer Ingelheim, Fitbit/Google, Medtronic, Premier, and IBM, and has consulted for Bristol Myers Squibb, Pfizer, Blackstone Life Sciences, and Invitae. A.A.P. is a Venture Partner at GV. He has received funding from IBM, Bayer, Pfizer, Microsoft, Verily, and Intel. C.U. serves on the Scientific Advisory Board of Immunai and Relation Therapeutics and has received sponsored research support from Janssen Pharmaceuticals. The remaining authors declare no competing interests.
