## [Peer review file · Nature Communications]

REVIEWER COMMENTS

Reviewer #1 (Remarks to the Author):

In the study, “Cross-Modal Autoencoder Framework Learns Holistic Representations of Cardiovascular State”, Radhakrishnan and colleagues present a strategy for learning signatures of cardiovascular health in unstructured data streams by combining information from two distinct domains. The benefit of such an approach is the development of meaningful learning from individual data streams, whereby one domain is specifically modeled to identify hidden information in another domain. Specifically, the study focuses on developing such an approach in UK Biobank, a richly characterized cohort study in the UK, using ECGs and cardiac MRI as the parallel data streams of interest. This is an interesting contribution with many potential applications. I have some comments for the authors’ consideration, to better clarify the value of their work.

1. The rationale for pursuing such a study is underdeveloped. In my review, I could see many potential applications of such cross-domain learning. The intended application also helps determine metrics for evaluation as well as possible alternative strategies. For readers of the Journal to appreciate the value of the contribution, it may be helpful to create a framework for why such learning may be meaningful - it will enhance the value of the work. Currently, 2 distinct tasks are being performed, (1) developing a cross-modality training framework, and how it can be used in generative tasks for a missing modality, and (2) and how this cross-modality learning can be used for genomic profiling. A setup in the introduction, and summary representation in the results, that focuses on the goals of the work could be further developed.
2. Another feature that likely would benefit from further development is the comparator of unimodal embeddings. Specifically, a unimodal autoencoder that is at best denoising the individual modalities and is trained on the reconstruction loss seems to be a little limited. Specifically, were the embedding for the AE optimized using structural phenotypic labels, e.g. demographic features, or structural cardiac parameters derived from the MRI (for ECG), or electrocardiographic measures (such as ECG intervals, for MRI), as opposed to contrastive cross-domain learning, wouldn’t the unimodal embeddings be more informative? I think that would be needed to prove the value of the specific contrastive learning approach, by demonstrating value beyond supervised learning on structured labels.
3. An admittedly fascinating takeaway from the work is the generative nature of the cross-modality embedding. However, the metrics to evaluate these generative models could be more meaningful. Specifically, if continuous measures are being predicted (on both ECG and MRI), a pairwise correlation coefficient (between generated and true representations) may allow more meaningful inference on the measures. This could replace the information presented in Figure S2, and potentially could be included in the main manuscript. Further, threshold based measures, such as LVEF and LVISD could be set at clinical thresholds to demonstrate the ability of the generated images to infer clinical phenotypes of interest (such as LV systolic dysfunction and LVH).

4. Similarly, if possible, a manual review of even a small subset of generated ECGs and/or MRIs for being meaningful representations of the originals may further improve the clinical utility of this work. If the authors have computed the measurements on cMRI and ECGs using manual review, that would be sufficient but wasn't clear. It may be helpful to clarify how the measurements on ECGs and cMRI were obtained.

5. As a follow-up to point #1 above, an application focus may also allow the methods to be more easily followed. As currently written, the strategic value of the results is missed. Specifically, a supervised training task that followed the pretraining is mentioned in the results, but the value of the clinical problem, and whether such cross-domain learning was actually essential is needed.

6. The iterative null space projection (INLP) being used as a strategy to address confounders in the GWAS inference should be further developed. Since these are different domains, specifically inferring genotypic representations from phenotypic embeddings, rather than reported characteristics, the mention of word embeddings about sex and race features is not clear. Page 4, "To remove the effect of possible confounders, we use the iterated nullspace projection (INLP) method [38], which was used in natural language processing for removing features such as race or gender information from word embeddings."

7. As a follow-up to #6, the challenge with confounders and their association with genotypes is unclear. Specifically, if a specific genotypic profile is being predicted from a different domain (ECG and MRI), it is certainly conceivable that some demographic and other confounding features are detected by these domains, which together lead to the identification of the genotypic variants for a given SNP. It is not clear in the study that the reason for the confounder adjustment is the interest in finding a true phenotypic representation of a genotype. If that is the case, then an embedding drawn from the data makes it challenging to address this directly without focusing on interpretability strategies such as saliency maps or other direct visualization. Therefore, the goal of identifying genotypic profiles from a cross-modal representation could be better set up, as the goals of the inference would determine the nature of confounder adjustment needed.

8. Figure 2A compares the embeddings of the unimodal ECG and MRI models vs the cross-modal ECG and MRI models is not clear. If the unimodal ECG and MRI use distinct model architectures, even if the size of the embedding is the same, it is unclear how they are being projected onto the same tSNE plot. Is there a strategy being applied to somehow standardize them - it will be helpful to lay out how that would be accomplished.

9. I believe the discussion section would benefit from further development. I think specifically highlighting the potential utility of learning cross-modal representations, in clinical and research domains may be helpful. Specifically, the community of investigators who will appreciate this research is those engaged in multimodal machine learning. The authors should develop why a genotype prediction strategy that uses available information using a mixture of experts or another strategy would be less useful than their approach. Moreover, the limitations of their approach deserve a clear enumeration. These limitations would also draw from the intended application.

Reviewer #2 (Remarks to the Author):

This paper proposes an autoencoder based framework for learning representations from multi-modal data. Specifically, the authors consider two modalities: MRIs and ECGs. The learned representations are applied for phenotype prediction and imputation of MRIs from ECGs.

The major weaknesses of this work are as follows.

1. The proposed method lack novelty. It is a combination of previous ideas including contrastive learning and data reconstruction, both of which have been broadly investigated for representation learning.

2. There are plenty of works for learning representations from multi-modal data (see below). The authors didn't compare with any of these baselines. As a result, it is difficult to assess the effectiveness of their autoencoder based method.

[1] Liu et al. Incomplete multi-modal representation learning for Alzheimer's disease diagnosis, 2021.

[2] Li et al. A survey of multi-view representation learning, 2018.

[3] Liang et al. Mind the gap: Understanding the modality gap in multi-modal contrastive representation learning, 2022.

[4] Ning et al. Relation-induced multi-modal shared representation learning for Alzheimer's disease diagnosis, 2021.

[5] Zhou et al. Deep multi-modal latent representation learning for automated dementia diagnosis, 2019.

3. The motivation of this study -- "Unlike these prior works that focus primarily on integrating images and vectorized data such as gene expression, we aim to integrate complex modalities with a temporal element (cardiac MRI videos and ECGs)" -- mentioned by the authors is unconvincing. Extending previous multi-modal representation learning methods to MRI videos and ECGs are straightforward: in these methods, one can simply replace encoders of images and vectorized data to encoders of videos and time-series data. There are no fundamental technical challenges in doing this. I suggest the authors to give more convincing justification on their study.

4. Since the authors position this work as multi-modal representation learning, at least three modalities should be used to demonstrate the effectiveness of the proposed method. Currently, only two modalities are used, which make the experiments unconvincing.

5. There is no external evaluation of the proposed method. Given the multi-modal representation learning model trained on UK Bank, does it generalize well on patient data in other hospitals?

6. In Figure 2c, there are no error bars. It is difficult to assess whether the improvement over baselines is statistically significant.

Overall, this work needs substantial improvement and revision.

Reviewer #3 (Remarks to the Author):

In their manuscript „Cross-modal autoencoder framework learn holistic representations of cardiovascular state“ by Radhakrishnan et al. the authors constructed a cross-modal computational (autoencoder/decoder) workflow from MRI and ECG data to improve the phenotype prediction from i) a single dataset (e.g. ECG), ii) MRI data imputed from ESC data and vice versa, iii) create a workflow for GWAS using phenotypes derived from the cross-model models. Clinical and phenotypical data were derived from the UKBiobank.

The authors show that the workflow

- could derive MRI phenotypes from ECG data only,
- could derive ECG phenotypes from MRI data only (less accurate than MRI from ECG),
- could predict general phenotypes such as gender and age from either ECG or MRI

that by using these computational derived MRI/ECG phenotypes, genome wide associations studies resulted in lead SNPs that were identified in previous GWAS analyses using the „real“ clinical phenotypes.

Thus, the authors conclude that MRI phenotypes can be imputed from SCG data only. Further their cross-modal workflow improved prediction of cardiovascular phenotypes by ECG data only, making this approach very useful as ECG phenotype data are easy to acquire compared to MRI and thus can derive much more information. Furthermore, by using this computational model to derive clinical phenotypes the authors showed that also unsupervised GWAS can be performed.

Also this reviewer is a non-computational scientist, this manuscript is very interesting and is good to read for someone not an expert in this field. Still, some questions and comments came up during reviewing the manuscript that should be adapted accordingly.

1. The authors used MRI and ECG data from the UKBB. Both approaches - in their conduction as well as in the evaluation of the images, are prone to inter-operator variability. How was this taken into account in the present manuscript? I assume that for research purposes, this variability can be reduced in selection the most appropriate data sets, however, for clinical application this needs to be taken into account, please also comment on the steps needed to translate your findings into clinical application.
2. In the cardiovascular field, some research focuses on circulating biomarkers (such as NTproBNP, Troponin, CRP) that can inform about clinical phenotypes. In my opinion, the authors should add biomarker as a „third level“ into their analyses and to see, whether ECG plus biomarkers can even better predict MRI phenotypes and general clinical phenotypes. This would extend the current clinical application procedures.
3. So far, the authors did not focus on a specific clinical disease (such as Arrhythmia/AF, or HF) but the structural and electrical features underlying these diseases. Would it be possible at this stage to already apply your model to diseased subjects to test whether the clinical diagnosis can indeed be improved?
4. Analyses performed included all subjects and were not stratified according to sex. I recommend to perform the analyses stratified by sex to see whether the models improve. Stratifications would also be interesting e.g for different BMI groups
5. In this regard: the analyses to derive general (categorical and conti) clinical phenotypes was performed from EITHER MRI or ECG data. How was decided which clinical measure was used, and were there difference observed when using MRI or ECG data?
6. I assume that the cross-modal tool will be applied to further phenotypes. It would be useful to see if related phenotypes/diseases such as neurological phenotypes can also be predicted. This should at least be discussed in more detail.
7. Genome –wide associations: Although provided in the Supplementary tables, I suggest to add an overview about the overlapping and distinct SNPs/loci that were identified by normal GWAS and the cross-modal GWAs, i.e. how many loci had been identified so far for a specific ECG phenotype with both methods, how many are overlapping, how many were not identified by the multi-modal approach? This will provide a better impression if the multi-modal approach is really comparable to the normal GWAS approach. In my opinion, this would be important, as it is not only interesting to see if SNPs are identified but further to establish the role and function of these SNPs or to generate a polygenetic risk score. If some loci/SNPs would be missed , important information will be lost.

8. UKBB age range: the age range of the subjects used is between 40-69, which does not represent the general population. Please comment on this in the discussion.

9. MRI Data: only the 4 chamber long axis has been used. Was the reason for this due to technical reasons (as these data were available in all subjects at best quality) or due to a computation reason? Please comment on this and include a statement how to integrate further (more complex?) features.

10. Were the results based on the multi-modal approach confirmed by a clinician, or was it confirmed by the ECG/MRI data itself? i.e. I assume that in some cases, the results were not 100% similar between MRI features and MRI-imputed features. How was this solved?

Reviewer #1 response:

In the study, “Cross-Modal Autoencoder Framework Learns Holistic Representations of Cardiovascular State”, Radhakrishnan and colleagues present a strategy for learning signatures of cardiovascular health in unstructured data streams by combining information from two distinct domains. The benefit of such an approach is the development of meaningful learning from individual data streams, whereby one domain is specifically modeled to identify hidden information in another domain. Specifically, the study focuses on developing such an approach in UK Biobank, a richly characterized cohort study in the UK, using ECGs and cardiac MRI as the parallel data streams of interest. This is an interesting contribution with many potential applications. I have some comments for the authors’ consideration, to better clarify the value of their work.

We are glad the reviewer found our work interesting and impactful for many applications. We hope our revision and responses below will help to further clarify the value of our work.

1. The rationale for pursuing such a study is underdeveloped. In my review, I could see many potential applications of such cross-domain learning. The intended application also helps determine metrics for evaluation as well as possible alternative strategies. For readers of the Journal to appreciate the value of the contribution, it may be helpful to create a framework for why such learning may be meaningful - it will enhance the value of the work. Currently, 2 distinct tasks are being performed, (1) developing a cross-modality training framework, and how it can be used in generative tasks for a missing modality, and (2) and how this cross-modality learning can be used for genomic profiling. A setup in the introduction, and summary representation in the results, that focuses on the goals of the work could be further developed.

We thank the reviewer for this suggestion and have added the following to the main text to provide a more detailed rationale for our study. In particular, per the reviewer’s suggestion, we have added text to the introduction and a summary at the beginning of the results section to emphasize the goals of this work.

Added to Introduction (Paragraph 1): “In particular, such cross-modal representations provide an opportunity for a broad range of downstream tasks such as (1) prediction of clinical phenotypes for diagnostics; (2) imputation of missing modalities in biomedical data; and (3) identification of genetic variants associated with a given organ system. Using the heart as a model system, we here develop such an integrative framework and show its effectiveness in these three downstream tasks.”

Added to Introduction (Paragraph 2): “Unlike these prior works that focus primarily on improving a specific downstream task such as phenotype prediction or modality translation through multi-modal data integration, we develop a generalized representation that improves performance on several downstream applications. We demonstrate the utility of this representation on three important biomedical tasks: in addition to phenotype prediction and multi-modal data integration and translation, we show that our cross-modal representation yields a new framework for characterizing genotype-phenotype associations. [...] Instead, our approach can identify SNPs that affect cardiac physiology in an unsupervised and general way. Namely, rather than merely identifying SNPs that affect a single phenotype such as the QT interval, our approach identifies SNPs that generally impact phenotypes present on ECGs or cardiac MRIs.”

Added to beginning of Results: “Overview of Results. We develop a cross-modal autoencoder framework to integrate ECG and cardiac MRI data from the UK Biobank [1]. We then leverage the resulting cross-modal embeddings to (1) improve prediction of clinical phenotypes; (2) enable modality translation between ECG and cardiac MRI; and (3) identify genetic variants that are associated with the cardiovascular system without requiring any labelled data.”

2. Another feature that likely would benefit from further development is the comparator of unimodal embeddings. Specifically, a unimodal autoencoder that is at best denoising the individual modalities and is trained on the reconstruction loss seems to be a little limited. Specifically, were the embedding for the AE optimized using structural phenotypic labels, e.g. demographic features, or structural cardiac parameters derived from the MRI (for ECG), or electrocardiographic measures (such as ECG intervals, for MRI), as opposed to contrastive cross-domain learning, wouldn’t the unimodal embeddings be more informative? I think that would be needed to prove the value of the specific contrastive learning approach,

by demonstrating value beyond supervised learning on structured labels.

We thank the reviewer for this suggestion. We have performed the suggested experiment, and the results are presented in Fig. 1 below. They demonstrate that prediction from cross-modal autoencoder embeddings outperforms prediction from a *semi-supervised autoencoder*, i.e., a unimodal autoencoder trained to both reconstruct a given modality and predict labels. This result matches intuition since prediction of clinical phenotypes and MRI-derived phenotypes is best from MRI embeddings, and our cross-modal embeddings are trained to integrate ECGs and MRIs.

We added this figure as Supplementary Fig. S3 in the revised manuscript and described it as follows in the Results section (paragraph 3): “In Supplementary Fig. S3, we additionally demonstrate that cross-modal representations outperform *semi-supervised* unimodal autoencoders, i.e., those trained to simultaneously autoencode and predict labels from the latent space.”

Figure 1: (Supplementary Fig. S3 in the revised manuscript) Prediction from cross-modal ECG embedding outperforms prediction from unimodal ECG embedding and semi-supervised ECG embedding, i.e., the embedding obtained from an autoencoder that is trained to both reconstruct ECGs and predict phenotypes.

3. An admittedly fascinating takeaway from the work is the generative nature of the cross-modality embedding. However, the metrics to evaluate these generative models could be more meaningful. Specifically, if continuous measures are being predicted (on both ECG and MRI), a pairwise correlation coefficient (between generated and true representations) may allow more meaningful inference on the measures. This could replace the information presented in Figure S2, and potentially could be included in the main manuscript. Further, threshold based measures, such as LVEF and LVISD could be set at clinical thresholds to demonstrate the ability of the generated images to infer clinical phenotypes of interest (such as LV systolic dysfunction and LVH).

We thank the reviewer for this comment. Since small translations of pixels can lead to a low pairwise correlation coefficient between generated and true MRIs, we felt that it would be more meaningful to compare the pairwise correlation coefficients of *phenotypes predicted from generated and true MRIs and ECGs*. An example of such an analysis was given in Fig. 3c of our manuscript. We thank the reviewer for the suggestion regarding adding an additional analysis using threshold based measures. We performed this analysis (see Fig 2 below); the results show again the benefit of using our cross-modal framework as compared to a unimodal or supervised framework. We added this figure as Supplementary Fig. S5 to

the revised manuscript and added the following text to the Methods section to describe these results.

“Prediction of left ventricular hypertrophy and left ventricular systolic dysfunction. We used LVM to derive thresholds for Left Ventricular Hypertrophy (LVH) and LVEF to derive thresholds for Left Ventricular Systolic Dysfunction (LVSD). To provide a binarized label for LVH, we first normalized all LVM measurements by dividing by body surface area (derived using the Mosteller method). We then stratified by sex and set the LVH label to be 1 if the normalized LVM was greater than 72 if the sex was male, respectively 1 if the normalized LVM was greater than 55 if the sex was female [2]. Supplementary Fig. S5 shows that using logistic regression from cross-modal embeddings leads to the highest AUROC of 0.756 for predicting LVH. For LVSD, the binarized label was obtained as an indicator of whether the LVEF was less than 45%. Again, logistic regression from cross-modal embeddings leads to the highest AUROC of 0.572. In both analyses, standard deviations were computed over 10-fold cross-validation.”

	Cross-modal ECG	Unimodal ECG	Supervised ECG
LVH AUROC	0.756 ± 0.022	0.716 ± 0.012	0.692 ± 0.016
LVSD AUROC	0.572 ± 0.052	0.535 ± 0.028	0.558 ± 0.023

Figure 2: (Supplementary Fig. S5 in the revised manuscript) Logistic regression using cross-modal ECG embeddings leads to improved prediction of Left Ventricular Hypertrophy (LVH) and Left Ventricular Systolic Dysfunction (LVSD) over logistic regression from unimodal ECG embeddings and supervised learning from ECGs directly.

4. Similarly, if possible, a manual review of even a small subset of generated ECGs and/or MRIs for being meaningful representations of the originals may further improve the clinical utility of this work. If the authors have computed the measurements on cMRI and ECGs using manual review, that would be sufficient but wasn’t clear. It may be helpful to clarify how the measurements on ECGs and cMRI were obtained.

In our experiments, we used the ECG and cMRI measurements provided directly by the UK Biobank [1] for validating the generated ECGs and cMRIs. In particular, in Fig. 2 in our manuscript, we trained models to predict these provided measurements from the generated ECGs and cMRIs and then evaluated the performance of these predictions on a held out test set. We clarified this by adding the following sentence in the Results section: “All MRI- and ECG-derived phenotypes as well as the categorical and continuous-valued physiological phenotypes were obtained from the UK Biobank.”

In addition, we evaluated the quality of the generated modalities by verifying that decoding translations in the latent space leads to interpretable shifts in the original modalities. For example, in Supplementary Fig. S10a, we showed that generating ECGs from embeddings shifted along the latent space direction of increasing QT interval leads to a corresponding increase in QT interval in the generated ECG. Similarly, in Fig. 3 and Supplementary Fig. S10b, we showed that the generated cMRIs accurately reflect changes upon increasing or decreasing LVM, RVEDV, and BMI.

5. As a follow-up to point #1 above, an application focus may also allow the methods to be more easily followed. As currently written, the strategic value of the results is missed. Specifically, a supervised training task that followed the pretraining is mentioned in the results, but the value of the clinical problem, and whether such cross-domain learning was actually essential is needed.

We thank the reviewer for this suggestion. To further emphasize the value of supervised learning from pretrained embeddings for clinical problems, we added the following sentence to the Results section summarizing some of our results: “Our results demonstrate the value of cross-modal embeddings for improving the prediction of clinical phenotypes including diseases such as left ventricular hypertrophy (LVH), left ventricular systolic dysfunction (LVSD), and hypercholesterolemia..”

Another key contribution of our work is improving the representational power of inexpensive ECGs for predicting clinical phenotypes by leveraging few cMRI samples. To emphasize this contribution, we added the following sentence to the Results section: “Overall, our cross-modal embeddings improve the representational power of inexpensive and prevalent ECGs for predicting clinical phenotypes by leveraging just a few MRI samples.”

6. The iterative null space projection (INLP) being used as a strategy to address confounders in the GWAS inference should be further developed. Since these are different domains, specifically inferring genotypic representations from phenotypic embeddings, rather than reported characteristics, the mention of word embeddings about sex and race features is not clear. Page 4, “To remove the effect of possible confounders, we use the iterated nullspace projection (INLP) method [38], which was used in natural language processing for removing features such as race or gender information from word embeddings.”

We thank the reviewer for this suggestion. To further clarify the INLP strategy for removing confounders, we added the following to the main text in the results section: “To remove the effect of possible confounders, we use the iterated nullspace projection (INLP) method [38]. This method was originally developed in the natural language processing domain to protect against biases (such as gender stereotypes) from appearing in word embeddings. We leverage INLP in the medical domain to ensure that confounders such as principal components of ancestry cannot be easily predicted from cross-modal embeddings, thereby ensuring that such features do not arise as confounders in the GWAS.”

7. As a follow-up to #6, the challenge with confounders and their association with genotypes is unclear. Specifically, if a specific genotypic profile is being predicted from a different domain (ECG and MRI), it is certainly conceivable that some demographic and other confounding features are detected by these domains, which together lead to the identification of the genotypic variants for a given SNP. It is not clear in the study that the reason for the confounder adjustment is the interest in finding a true phenotypic representation of a genotype. If that is the case, then an embedding drawn from the data makes it challenging to address this directly without focusing on interpretability strategies such as saliency maps or other direct visualization. Therefore, the goal of identifying genotypic profiles from a cross-modal representation could be better set up, as the goals of the inference would determine the nature of confounder adjustment needed.

The challenge with confounders is that they are so easily predicted from latent embeddings that they can hide almost all genetic signal, as is shown in Supplementary Fig. S13b. To emphasize this point we added the following sentence to the Results section: “Without accounting for confounders, MANOVA is unable to recover genetic signal from the latent embeddings (see Supplementary Fig. S13a,b).”

Indeed, our Fig. 2c demonstrates that confounders such as sex can be almost perfectly identified from embeddings. This was a key motivation for utilizing INLP to remove such confounders from the data. An important finding of our work is that such adjustment was sufficient for identifying the majority of genes associated with a cardiovascular phenotype, as is shown in Fig. 4b and Supplementary Fig. S12.

While interpretability strategies such as saliency maps or direct visualization can be useful for identifying the phenotypic impact of a given SNP, these require manual review and would be incredibly time-intensive given the number of SNPs. On the other hand, our Fig. 4b and d demonstrate that we can automatically identify and cluster SNPs by their phenotypic effect, thereby drastically streamlining this process. To emphasize this point, we added the following text to the Results section of the revised manuscript: “While we so far demonstrated that we can perform GWAS on one-dimensional traits using our cross-modal embedding, we note two limitations of this approach. The first limitation is that genetic variants are often pleiotropic, affecting many traits at the same time. Indeed, a variant can have a small effect on a pair of traits and looking at one trait alone would not provide sufficient power for identifying this variant. The second limitation is that one-dimensional traits, even in aggregate, are an incomplete characterization of the information in a diagnostic modality. For example, in ECGs, we aim to not only identify variants that impact the measured phenotypes such as the QT interval but variants that affect the ECG in any way. In the following, we therefore develop an unsupervised GWAS methodology that provides a principled approach for automatically discovering genetic variants from rich diagnostic modalities without using any labelled phenotype measurements or turning to manual, time-intensive interpretability strategies such as saliency maps or direct visualization of ECGs and MRIs.”

8. Figure 2A compares the embeddings of the unimodal ECG and MRI models vs the cross-modal ECG and MRI models is not clear. If the unimodal ECG and MRI use distinct model architectures, even if the size of the embedding is the same, it is unclear how they are being projected onto the same tSNE plot. Is there a strategy being applied to somehow standardize them - it will be helpful to lay out how that would be accomplished.

To generate Fig. 2a, we simply performed t-SNE on the ECG and MRI embeddings obtained from the corresponding unimodal autoencoders. These embeddings all have the same number of dimensions (256) by construction. The key message of this figure is that, unsurprisingly, there is no guarantee of alignment between the two modalities without coupling them with a cross-modal autoencoder. We

added the following sentence to the Results section to clarify this point: “We can use a combined t-SNE visualization of the two modalities also for the latent space embedding obtained from the unimodal autoencoders, since our ECG and MRI autoencoders both use latent embeddings of the same size (256 dimensions).”

9. I believe the discussion section would benefit from further development. I think specifically highlighting the potential utility of learning cross-modal representations, in clinical and research domains may be helpful. Specifically, the community of investigators who will appreciate this research is those engaged in multimodal machine learning. The authors should develop why a genotype prediction strategy that uses available information using a mixture of experts or another strategy would be less useful than their approach. Moreover, the limitations of their approach deserve a clear enumeration. These limitations would also draw from the intended application.

We thank the reviewer for these helpful suggestions. We added the following text to the Discussion section to (1) further highlight the utility of cross-modal representations; (2) clarify the benefit of our method for genotype prediction over alternative strategies relying on labelled data such as mixture-of-experts methods; and (3) further clarify the limitations of our approach:

“The reliable performance boost in phenotype prediction from a cross-modal embedding further highlights its applicability to aid in diagnostics. An interesting future application of our framework is to determine the extent to which cross-modal ECG embeddings can be translated to a hospital setting and, in conjunction with other biomarkers, improve the prediction of specific cardiovascular diseases.”

“The proposed unsupervised GWAS method provides an effective and efficient approach to genetic discovery as it has the same computational cost as performing a single GWAS and, in contrast to existing methods for GWAS, it does not require any labelled phenotype data.”

“However, we acknowledge that a current limitation of our work is that UK Biobank samples are limited in their diversity with individuals primarily falling between the ages of 40 to 69. In addition, the UK Biobank is known to contain racial and socioeconomic biases, which can lead to problematic inequities in terms of healthcare [3]. It would therefore be important to re-train or update our model on a more diverse population and perform a careful analysis of how well our model generalizes to underrepresented cohorts in future work before translating this method to hospital settings.”

Reviewer #2 Response:

This paper proposes an autoencoder based framework for learning representations from multi-modal data. Specifically, the authors consider two modalities: MRIs and ECGs. The learned representations are applied for phenotype prediction and imputation of MRIs from ECGs.

We thank the reviewer for the comments. We would like to highlight an additional key novelty of our work, namely the development of a method to enable performing unsupervised GWAS using the cross-modal latent space. We feel this is a key novelty in our work: as far as we are aware, our work is the first to present a general, computationally efficient approach to performing GWAS from any given modality without needing labelled phenotype data.

The major weaknesses of this work are as follows.

1. The proposed method lack novelty. It is a combination of previous ideas including contrastive learning and data reconstruction, both of which have been broadly investigated for representation learning.

As described above, a key novelty of our work is that we develop a novel method for performing unsupervised GWAS from the cross-modal embedding. In addition, we would like to point out that the proposed framework for integrating ECGs and cardiac MRIs is a novel application of cross-modal autoencoders and contrastive learning. As also remarked by the reviewer in the following point, the literature on multi-modal learning is vast; this is due to the fact that this field is complex and different methods are needed for different data modalities and different downstream tasks.

2. There are plenty of works for learning representations from multi-modal data (see below). The authors didn't compare with any of these baselines. As a result, it is difficult to assess the effectiveness of their autoencoder based method.

1. Liu et al. Incomplete multi-modal representation learning for Alzheimer's disease diagnosis, 2021.
2. Li et al. A survey of multi-view representation learning, 2018.
3. Liang et al. Mind the gap: Understanding the modality gap in multi-modal contrastive representation learning, 2022.
4. Ning et al. Relation-induced multi-modal shared representation learning for Alzheimer's disease diagnosis, 2021.
5. Zhou et al. Deep multi-modal latent representation learning for automated dementia diagnosis, 2019.

We would like to reiterate the importance of the downstream task for multi-modal data integration methods. None of the above works develop techniques to use the latent embedding to perform a GWAS and thus cannot be used as a point of comparison with our work. Also with regard to phenotype prediction the above papers differ from ours in their aim: a major goal of our work is to boost the predictive performance of a single data modality that is highly prevalent (namely ECG data) by using only few samples from both modalities (ECG and the much more costly cardiac MRI samples). This is in stark contrast to the works referenced by the reviewer, which aim to boost performance of disease prediction (such as Alzheimer's) from all modalities together. In the following, we provide a brief summary of each of these prior works and highlight key differences to our work in more detail.

Liu et al. Incomplete multi-modal representation learning for Alzheimer's disease diagnosis, 2021. This work presents a multi-modal integration framework to improve prediction of Alzheimer's disease. Namely, this work utilizes autoencoders to learn latent representations of features extracted from PET and MRI modalities, develops a method to impute missing modalities based on kernel matrices from latent representations, and uses a kernel canonical correlation analysis (CCA) to learn a common representation for predicting the disease. The scope of this work is mainly the prediction of disease from an integrated representation of two data modalities. In contrast to our work, their framework cannot directly be applied to translate between modalities. In addition, they do not consider the problem of performing GWAS from the latent representation or improving prediction of disease from a single modality resulting from a cross-modal representation obtained from few paired samples of different modalities.

Li et al. A survey of multi-view representation learning, 2018. This work presents a survey of a variety of multi-modal representation learning approaches, including CCA and fusion with multi-modal autoencoders. The methods reviewed in this work are either primarily focused on downstream prediction tasks using the multimodal embedding or translating between embeddings/data modalities. This survey does not contain methods for performing GWAS from latent representations. In addition, the applications considered in this survey are primarily in computer vision and natural language processing and are far removed from the healthcare applications considered in our work.

Liang et al. Mind the gap: Understanding the modality gap in multi-modal contrastive representation learning, 2022. This paper demonstrates that modern contrastive learning models (such as CLIP [4]) that integrate text and image data result in embeddings that separate both image and text data by a fixed distance. This so-called modality gap is a result of model initialization and persists through contrastive learning and depends on hyper-parameters such as temperature. The paper also demonstrates that adjusting the size of this gap can have marginal effects on downstream tasks such as evaluating model biases. This work is a study of a phenomenon arising in existing contrastive learning models applied to image and text data. Moreover, this work does not consider any models used in a healthcare settings, nor does it consider the task of translating between different data modalities or performing a GWAS analysis in the latent space.

Ning et al. Relation-induced multi-modal shared representation learning for Alzheimer’s disease diagnosis, 2021. This work develops a matrix factorization based framework for integrating features extracted from PET and MRI in order to predict Alzheimer’s disease. This work does not use modality specific convolutional autoencoders but rather works with matrix factorization on features extracted from the modalities. As such, it cannot be used for translating between modalities. In addition, this work does not consider the problem of performing GWAS. Furthermore, this work is primarily focused on the prediction of Alzheimer’s disease from all modalities rather than boosting the predictive performance from a single, easily accessible modality using only few multi-modal samples.

Zhou et al. Deep multi-modal latent representation learning for automated dementia diagnosis, 2019. This work presents a deep non-negative matrix factorization approach for integrating features extracted from PET and MRI in order to predict varying levels of cognitive impairment. As the previous paper, this work does not use modality specific convolutional autoencoders but rather works with matrix factorization on features extracted from the modalities. As such, it cannot be used for translating between modalities. In addition, this work does not consider the problem of identifying genetic variants associated with cognitive impairment. Furthermore, this work is primarily focused on the prediction of disease from all modalities rather than improving predictive performance from a single, easily accessible modality using only few multi-modal samples.

We thank the reviewer for providing these references, which we now added in the Introduction of the revised manuscript. We also added a summary of the key differences to prior work, highlighting the novelty of our work: “Multi-modal data integration is a rich field with a variety of methods developed for specific applications. A survey of multi-modal approaches is presented in [5]. Unlike multi-modal data integration approaches based on classical methods such as canonical correlation analysis (CCA) [6–9] or non-negative matrix factorization [10, 11], our approach relies on a class of machine learning models called autoencoders [...] Unlike these prior works that focus primarily on improving a specific downstream task such as phenotype prediction or modality translation through multi-modal data integration, we develop a generalized representation that improves performance on several downstream applications. We demonstrate the utility of this representation on three important biomedical tasks: in addition to phenotype prediction and multi-modal data integration and translation, we show that our cross-modal representation yields a new framework for characterizing genotype-phenotype associations.”

3. The motivation of this study – “Unlike these prior works that focus primarily on integrating images and vectorized data such as gene expression, we aim to integrate complex modalities with a temporal element (cardiac MRI videos and ECGs)” – mentioned by the authors is unconvincing. Extending previous multi-modal representation learning methods to MRI videos and ECGs are straightforward: in these methods, one can simply replace encoders of images and vectorized data to encoders of videos and time-series data. There are no fundamental technical challenges in doing this. I suggest the authors to give more convincing justification on their study.

The sentence referenced by the reviewer should be considered within its context. One of the aims of our work is to integrate complex modalities with a temporal element: we mentioned this to contrast our work from previous work. The computational challenges of training large-scale models to embed these

more complex modalities that have a temporal element such as cardiac MRI videos and ECGs should not be underestimated. But the main goal of our work is to provide a general framework for integrating modalities that can be used for three downstream tasks, namely: (1) improve phenotype prediction from the joint embedding; (2) enable translation between modalities; and (3) simplify the discovery of genetic variants. To clarify the rationale for our work, we added the following text in the revised manuscript.

Added to Introduction (paragraph 1): “In particular, such cross-modal representations provide an opportunity for a broad range of downstream tasks such as (1) prediction of clinical phenotypes for diagnostics; (2) imputation of missing modalities in biomedical data; and (3) identification of genetic variants associated with a given organ system. Using the heart as a model system, we here develop such an integrative framework and show its effectiveness in these three downstream tasks.”

Added to Introduction (paragraph 2): “Unlike these prior works that focus primarily on improving a specific downstream task such as phenotype prediction or modality translation through multi-modal data integration, we develop a generalized representation that improves performance on several downstream applications. We demonstrate the utility of this representation on three important biomedical tasks: in addition to phenotype prediction and multi-modal data integration and translation, we show that our cross-modal representation yields a new framework for characterizing genotype-phenotype associations. While various prior works have conducted genome-wide association studies (GWAS) to identify single nucleotide polymorphisms (SNPs) associated with cardiovascular diseases [12, 13], features measured on ECGs [14, 15], or features measured on cardiac MRI [16, 17], these GWAS approaches have relied on labelled data derived from individual modalities. Instead, our approach can identify SNPs that affect cardiac physiology in an unsupervised and general way. Namely, rather than merely identifying SNPs that affect a single phenotype such as the QT interval, our approach identifies SNPs that generally impact phenotypes present on ECGs or cardiac MRIs.”

Added to beginning of the Results section: “Overview of Results. We develop a cross-modal autoencoder framework to integrate ECG and cardiac MRI data from the UK Biobank [1]. We then leverage the resulting cross-modal embeddings to (1) improve prediction of clinical phenotypes; (2) enable modality translation between ECG and cardiac MRI; and (3) identify genetic variants that are associated with the cardiovascular system without requiring any labelled data.”

4. Since the authors position this work as multi-modal representation learning, at least three modalities should be used to demonstrate the effectiveness of the proposed method. Currently, only two modalities are used, which make the experiments unconvincing.

As suggested by the reviewer, we performed an additional analysis, where we integrated three modalities, namely: (1) ECGs; (2) long axis views of cardiac MRIs; and (3) short axis views of cardiac MRIs. This analysis demonstrates that our framework can easily be used when more than two modalities are available. The results of this analysis are shown in Fig. 3 below. It shows that integrating a short axis view of cardiac MRIs provides a boost to the prediction of general phenotypes such as BMI or sex, but, as expected, reduces the predictive power on other phenotypes that are hard to predict from the short axis view alone. For example, the prediction of intervals derived from ECGs worsens when integrating short axis views, which is expected since the short axis view provides little information on this phenotype and may in fact add noise to the prediction task. We added Fig. 3 as Supplementary Fig. S2 in the revised manuscript, and we added the following sentence to reference this analysis in the main text: “While we mainly apply our framework to integrate two modalities (ECGs and cardiac MRIs), we demonstrate that it can also be applied to three or more modalities in Supplementary Fig. S2.”

Figure 3: (Supplementary Fig. S2 in the revised manuscript) A comparison of phenotype prediction from bi-modal embeddings of ECGs and long axis views of cardiac MRIs (LAX) and tri-modal embeddings of ECGs, LAX, and short axis view of cardiac MRIs (SAX). We observe that predictive performance of general phenotypes such as BMI and sex improves when using tri-modal embeddings since SAX is informative about these phenotypes. On the other hand, prediction of ECG intervals (e.g., QT, RR, and PQ intervals) from tri-modal embeddings decreases since SAX contains little information about these phenotypes and may add noise to the prediction task.

5. There is no external evaluation of the proposed method. Given the multi-modal representation learning model trained on UK Bank, does it generalize well on patient data in other hospitals?

While collecting and analyzing data from an external cohort is outside the scope of our current work, we agree that an external evaluation on patient data in other hospitals is an important avenue for future work. To emphasize this point, we added the following text in the Discussion section: “the UK Biobank is known to contain racial and socioeconomic biases, which can lead to problematic inequities in terms of healthcare [3]. It would therefore be important to re-train or update our model on a more diverse population and perform a careful analysis of how well our model generalizes to underrepresented cohorts in future work before translating this method to hospital settings.”

6. In Figure 2c, there are no error bars. It is difficult to assess whether the improvement over baselines is statistically significant.

We thank the reviewer for this comment. To better demonstrate the effectiveness of our method over baselines, we added error bars to Figure 2c in the revised manuscript based on 5-fold cross-validation. The updated figure is shown below in Fig. 4. The error bars further demonstrate that prediction from cross-modal embeddings outperforms prediction from unimodal embeddings and standard supervised learning.

Figure 4: Improvement of phenotype prediction from cross-modal representations over unimodal representations or supervised learning from the original modalities. (a) A t-SNE visualization of the cross-modal embeddings for the ECG and MRI samples demonstrates that the modality specific embeddings are well-mixed, unlike the modality specific embeddings obtained from the unimodal autoencoders. (b) Ranking each MRI by its cosine similarity with a given ECG in the latent space, we visualize the accuracy that the ground truth MRI appears in the top k neighbors among 4752 test ECG-MRI pairs from the UK Biobank. (c) Kernel regression on cross-modal representations outperforms kernel regression on unimodal representations and supervised deep learning methods on 4 different tasks: (1) prediction of ECG derived phenotypes from MRIs only; (2) prediction of MRI derived phenotypes from ECG only; (3) prediction of general physiological phenotypes that are of categorical nature from either ECG or MRI; and (4) prediction of general physiological phenotypes that are of continuous nature from either ECG or MRI. All MRI phenotype abbreviations are defined in Methods 1.3. **Error bars are computed using 5-fold cross-validation.** (d) Analysis of the scaling law when utilizing our framework for predicting MRI derived phenotypes from ECGs only. We observe that increasing the number of unlabelled ECG-MRI pairs for pre-training boosts the mean R^2 prediction of 9 MRI derived phenotypes by twice as much as increasing the number of labelled MRI samples. This analysis highlights the benefit of collecting more unlabelled ECG-MRI pairs as compared to paired labelled examples for this task.

Reviewer #3 Response:

In their manuscript “Cross-modal autoencoder framework learn holistic representations of cardiovascular state” by Radhakrishnan et al. the authors constructed a cross-modal computational (autoencoder/decoder) workflow from MRI and ECG data to improve the phenotype prediction from i) a single dataset (e.g. ECG), ii) MRI data imputed from ECG data and vice versa, iii) create a workflow for GWAS using phenotypes derived from the cross-modal models. Clinical and phenotypical data were derived from the UKBiobank. The authors show that the workflow

- could derive MRI phenotypes from ECG data only,
- could derive ECG phenotypes from MRI data only (less accurate than MRI from ECG),
- could predict general phenotypes such as gender and age from either ECG or MRI

that by using these computational derived MRI/ECG phenotypes, genome wide associations studies resulted in lead SNPs that were identified in previous GWAS analyses using the “real” clinical phenotypes. Thus, the authors conclude that MRI phenotypes can be imputed from ECG data only. Further their cross-modal workflow improved prediction of cardiovascular phenotypes by ECG data only, making this approach very useful as ECG phenotype data are easy to acquire compared to MRI and thus can derive much more information. Furthermore, by using this computational model to derive clinical phenotypes the authors showed that also unsupervised GWAS can be performed.

Also this reviewer is a non-computational scientist, this manuscript is very interesting and is good to read for someone not an expert in this field. Still, some questions and comments came up during reviewing the manuscript that should be adapted accordingly.

We thank the reviewer for their positive comments and in particular, for finding our work interesting and a good read for non-experts in this field. We hope the following point-by-point response addresses the raised questions.

1. The authors used MRI and ECG data from the UKBB. Both approaches - in their conduction as well as in the evaluation of the images, are prone to inter-operator variability. How was this taken into account in the present manuscript? I assume that for research purposes, this variability can be reduced in selection the most appropriate data sets, however, for clinical application this needs to be taken into account, please also comment on the steps needed to translate your findings into clinical application.

We thank the reviewer for raising the question about inter-operator variability in human-based interpretation of ECGs and MRIs and during data collection. In this specific application, variability due to human-based interpretation is less of a concern, since the majority of phenotypes we consider (e.g. MRI-derived and ECG-derived phenotypes) are reported by the UK Biobank using fully automated methods. To reduce effects of inter-operator variability during data collection, we first restricted to those individuals who had ECGs and MRI data taken on the same day in a fixed assessment center. There are 38,686 such individuals. We additionally performed pre-processing of the data (described in Methods) to reduce noise in the data. An example is using median ECG wave-forms in order to reduce the effects of ECG drift. For future clinical applications, one option is to use transfer learning approaches to efficiently adapt our model to the variability present in other hospital data. We added the following sentence to the Methods section to emphasize that the measurements for each individual occur in the same assessment center: “The ECG and MRI data for an individual are collected in the same assessment center.”

2. In the cardiovascular field, some research focuses on circulating biomarkers (such as NTproBNP, Troponin, CRP) that can inform about clinical phenotypes. In my opinion, the authors should add biomarker as a “third level” into their analyses and to see, whether ECG plus biomarkers can even better predict MRI phenotypes and general clinical phenotypes. This would extend the current clinical application procedures.

We thank the reviewer for this suggestion. We performed an additional analysis to evaluate the impact of including circulating biomarkers when predicting MRI derived phenotypes or general clinical phenotypes. The results are shown in Fig. 5 below, which we added as Supplementary Fig. S6 to the revised manuscript. In particular, in the UK Biobank we had access to CRP and LDL but not to NTproBNP and Troponin. Utilizing CRP and LDL, we found that the predictive performance for BMI and age improved in all models, while the performance remained comparable for the prediction of MRI derived phenotypes, sex, hypertension, and hypercholesterolemia. We note that the prediction of hypercholesterolemia and hypertension may not have improved significantly since unexpectedly in the 2994 training samples LDL was negatively correlated with hypertension and hypercholesterolemia incidence rates: those with hypercholesterolemia had a mean LDL of 0.78 (standard deviation of 1.54) and

those without hypercholesterolemia had a mean LDL of 2.78 (standard deviation of 1.64); similarly, those with hypertension had a mean LDL of 1.22 (standard deviation of 1.75) and those without hypertension had a mean LDL of 2.35 (standard deviation of 1.83). While it is seemingly counter-intuitive that individuals with a diagnosis of hypercholesterolemia would have lower LDL, we presume this is because they are being treated with a lipid lowering medication.

We included this analysis in Supplementary Fig. S6 together with the following sentences in the Results section: "In addition, in Supplementary Fig. S6, we showcase the impact of incorporating circulating biomarkers such as C-reactive protein (CRP) and low-density lipoproteins (LDL) on phenotype prediction. In general, we find that CRP and LDL improve performance for predicting age and BMI."

	Without LDL & CRP			With LDL & CRP		
	Cross-modal	Unimodal	Supervised	Cross-modal	Unimodal	Supervised
LVM	0.536	0.475	0.439	0.536	0.485	0.440
LVEDV	0.451	0.382	0.381	0.442	0.379	0.383
LVEF	0.103	0.080	0.049	0.098	0.085	0.044
LVESV	0.380	0.324	0.327	0.368	0.325	0.326
LVSV	0.316	0.246	0.231	0.315	0.244	0.233
RVEF	0.129	0.116	0.065	0.129	0.120	0.063
RVESV	0.445	0.388	0.374	0.447	0.399	0.380
RVSV	0.320	0.245	0.236	0.320	0.248	0.239
RVEDV	0.490	0.409	0.407	0.490	0.418	0.413
Average	0.352	0.296	0.279	0.349	0.300	0.280

	Without LDL & CRP			With LDL & CRP		
	Cross-modal	Unimodal	Supervised	Cross-modal	Unimodal	Supervised
BMI	0.362	0.320	0.192	0.461	0.426	0.330
Age	0.264	0.253	0.105	0.278	0.253	0.117
Average	0.313	0.286	0.148	0.370	0.340	0.224

	Without LDL & CRP			With LDL & CRP		
	Cross-modal	Unimodal	Supervised	Cross-modal	Unimodal	Supervised
Sex	0.961	0.937	0.911	0.962	0.947	0.909
Hypercholesterolemia	0.635	0.629	0.598	0.630	0.625	0.572
Hypertension	0.696	0.713	0.684	0.706	0.703	0.685
Average	0.764	0.760	0.731	0.766	0.758	0.722

Figure 5: (Supplementary Fig. S6 in the revised manuscript) Impact of two circulating biomarkers, namely low-density lipoprotein (LDL) and C-reactive protein (CRP), on the prediction of phenotypes from ECG. Generally, we observe that the inclusion of these biomarkers consistently increases the R^2 -value for predicting BMI and age across all models but does not generally increase prediction accuracy for MRI derived phenotypes or sex, hypercholesterolemia, and hypertension.

3. So far, the authors did not focus on a specific clinical disease (such as Arrhythmia/AF, or HF) but the structural and electrical features underlying these diseases. Would it be possible at this stage to already apply your model to diseased subjects to test whether the clinical diagnosis can indeed be improved?

To apply our model for the prediction of a specific disease such as AF or HF, we need few labeled samples of patients with and without the disease. Using such data together with our framework, one can then build a predictor for a given disease in the multi-modal embedding. To provide preliminary evidence that our model can be used to improve AF or HF prediction, we show that the cross-modal representation improves the performance of predictive models trained using labels of AF and HF provided by the UK Biobank. These results are shown in Fig. 6 below (which we added as Supplementary Fig. S8 in the revised manuscript) and described in the Results section as follows: "Lastly, in Supplementary Fig. S8, we demonstrate that the cross-modal representation can improve prediction for diseases such as atrial fibrillation (AF) or heart failure (HF) using labels provided by the UK Biobank."

Figure 6: (Supplementary Fig. S8 in the revised manuscript) Cross-modal representation leads to improved performance of atrial fibrillation (AF) and heart failure (HF). Labels of AF and HF were provided by the UK Biobank.

We also added a further use-case of the cross-model framework in the revised manuscript, demon-

strating that it can be used to improve the prediction accuracy of Left Ventricular Hypertrophy (LVH) and Left Ventricular Systolic Dysfunction (LVSD). These results are shown in Fig. 2 (Supplementary Fig. S5 in the revised manuscript) and described in the Methods section of the revised manuscript as follows:

“Prediction of left ventricular hypertrophy and left ventricular systolic dysfunction. We used LVM to derive thresholds for Left Ventricular Hypertrophy (LVH) and LVEF to derive thresholds for Left Ventricular Systolic Dysfunction (LVSD). To provide a binarized label for LVH, we first normalized all LVM measurements by dividing by body surface area (derived using the Mosteller method). We then stratified by sex and set the LVH label to be 1 if the normalized LVM was greater than 72 if the sex was male, respectively 1 if the normalized LVM was greater than 55 if the sex was female [2]. Supplementary Fig. S5 shows that using logistic regression from cross-modal embeddings leads to the highest AUROC of 0.756 for predicting LVH. For LVSD, the binarized label was obtained as an indicator of whether the LVEF was less than 45%. Again, logistic regression from cross-modal embeddings leads to the highest AUROC of 0.572. In both analyses, standard deviations were computed over 10-fold cross-validation.”

We also added the following sentence to the Discussion section to clarify how only a few labelled samples are needed to fine-tune our framework for the effective prediction of different clinical phenotypes: “While we showed how our cross-modal embedding of ECG and cardiac MRI data can be used to improve the prediction of clinical phenotypes such as LVH, LVSD, and hypercholesterolemia, our framework is general and only requires a few labelled samples to be applicable to other clinical phenotypes.”

4. Analyses performed included all subjects and were not stratified according to sex. I recommend to perform the analyses stratified by sex to see whether the models improve. Stratification would also be interesting e.g for different BMI groups.

We thank the reviewer for this suggestion. We performed an additional analysis to evaluate the effect of stratification by sex and high vs. low BMI for predicting MRI derived phenotypes from ECG embeddings. The results are shown in Fig. 7 below and were added as Supplementary Fig. S7 in the revised manuscript. As suggested by the reviewer, we indeed observe that stratifying by sex and BMI leads to improved performance for all models. However, the use of cross-modal ECG embeddings still leads to the best prediction performance of MRI derived phenotypes for all models. This result was summarized as following in the Results section in the revised manuscript: “Furthermore, in Supplementary Fig. S7, we demonstrate that there is a boost in the prediction of MRI-derived phenotypes when stratifying phenotypes by sex and BMI.”

	Without Sex			With Sex		
	Cross-modal	Unimodal	Supervised	Cross-modal	Unimodal	Supervised
LVM	0.535	0.469	0.433	0.594	0.576	0.490
LVEDV	0.452	0.387	0.391	0.506	0.495	0.415
LVEF	0.109	0.095	0.049	0.123	0.112	0.052
LVESV	0.388	0.344	0.352	0.437	0.428	0.369
LVS	0.311	0.240	0.220	0.345	0.319	0.241
RVEF	0.131	0.111	0.061	0.146	0.140	0.073
RVESV	0.438	0.382	0.370	0.493	0.489	0.404
RVS	0.312	0.242	0.222	0.356	0.337	0.246
RVEDV	0.479	0.403	0.399	0.542	0.530	0.436
Average	0.351	0.297	0.277	0.394	0.381	0.303
	Without BMI			With BMI		
	Cross-modal	Unimodal	Supervised	Cross-modal	Unimodal	Supervised
LVM	0.535	0.469	0.433	0.566	0.513	0.475
LVEDV	0.452	0.388	0.391	0.471	0.428	0.400
LVEF	0.108	0.094	0.049	0.111	0.085	0.048
LVESV	0.388	0.344	0.352	0.401	0.359	0.346
LVS	0.312	0.242	0.222	0.328	0.286	0.241
RVEF	0.131	0.113	0.061	0.132	0.099	0.066
RVESV	0.439	0.382	0.370	0.445	0.398	0.367
RVS	0.313	0.244	0.223	0.340	0.296	0.250
RVEDV	0.480	0.403	0.400	0.499	0.446	0.409
Average	0.351	0.298	0.278	0.366	0.323	0.289

Figure 7: (Supplementary Fig. S7 in the revised manuscript) **Impact of stratification by sex and BMI on the prediction of MRI-derived phenotypes from ECG.** In general, we observe that such stratification increases the prediction accuracy as measured by R²-values.

5. In this regard: the analyses to derive general (categorical and conti) clinical phenotypes was performed from EITHER MRI or ECG data. How was decided which clinical measure was used, and were there difference observed when using MRI or ECG data?

Our focus was primarily on the prediction of phenotypes from ECG data alone, since ECGs are easier and less expensive to collect than cardiac MRIs. Nevertheless, for completeness, we also discussed the prediction of phenotypes from MRI data. The bottom row of Fig. 2c highlights the difference between prediction from ECG and MRI for general clinical phenotypes. In general, as expected, we observe that for these clinical phenotypes, prediction accuracy is the highest for cross-modal MRI. But importantly,

we observe that prediction from cross-modal ECG embeddings outperforms the prediction from unimodal ECG embeddings and supervised learning directly from ECG data. This has an important implication, namely that we are able to improve the representational power of ECGs and improve prediction of clinical phenotypes given little cardiac MRI data. To emphasize this point, we added the following sentence in the Results section of the revised manuscript: “Overall, our cross-modal embeddings improve the representational power of inexpensive and prevalent ECGs for predicting clinical phenotypes by leveraging just a few MRI samples.”.

6. I assume that the cross-modal tool will be applied to further phenotypes. It would be useful to see if related phenotypes/diseases such as neurological phenotypes can also be predicted. This should at least be discussed in more detail.

Indeed, our cross-modal framework is a general method that can be applied to predict other phenotypes including neurological diseases. While out of scope for the current work, we agree that it would be very interesting in future work to analyze the predictive performance of the cross-modal ECG and cardiac MRI embedding for other phenotypes/diseases including neurological phenotypes/diseases. We added the following sentences in the Discussion section to highlight this important future research direction.

“While we showed how our cross-modal embedding of ECG and cardiac MRI data can be used to improve the prediction of clinical phenotypes such as LVH, LVSD, and hypercholesterolemia, our framework is general and only requires a few labelled samples to be applicable to other clinical phenotypes. As such, an interesting direction for future research is to understand the extent to which related neurological phenotypes can be predicted from cross-modal ECG and MRI embeddings.”

7. Genome-wide associations: Although provided in the Supplementary tables, I suggest to add an overview about the overlapping and distinct SNPs/loci that were identified by normal GWAS and the cross-modal GWAS, i.e. how many loci had been identified so far for a specific ECG phenotype with both methods, how many are overlapping, how many were not identified by the multi-modal approach? This will provide a better impression if the multi-modal approach is really comparable to the normal GWAS approach. In my opinion, this would be important, as it is not only interesting to see if SNPs are identified but further to establish the role and function of these SNPs or to generate a polygenetic risk score. If some loci/SNPs would be missed, important information will be lost.

We thank the reviewer for this suggestion. We added Fig. 8 below (which is Supplementary Fig. S15 in the revised manuscript) as suggested, which demonstrates the difference in SNPs found by the cross-modal unsupervised GWAS and the SNPs found by the supervised GWAS on ECG-derived phenotypes. Overall, we note that the SNPs not found by our method are close to the significance cutoff of 5×10^{-8} (equivalent to a negative log p-value of 7.25). On the other hand, we note that the SNPs found by our method and supervised GWAS have high negative log p-values (e.g., for QRS duration, the maximum log p-value of SNPs found by both methods is 44.9887). We added the following text to the Results section to reference this figure: “In Supplementary Fig. S15, we provide Venn diagrams comparing the SNPs found by the unsupervised GWAS and those found by the supervised GWAS on ECG-derived phenotypes.”

Figure 8: (Supplementary Fig. S15 in the revised manuscript) Venn diagrams illustrating the difference in SNPs found by the cross-modal unsupervised GWAS and SNPs found by the supervised GWAS on ECG-derived phenotypes. Overall, we observe that the SNPs not found by our method are near the significance cutoff of 5×10^{-8} .

8. UKBB age range: the age range of the subjects used is between 40-69, which does not represent the general population. Please comment on this in the discussion.

We thank the reviewer for this important comment. We added the following sentences to the Discussion section to acknowledge the limitation in terms of diversity with respect to age, but also ethnicity and socioeconomic status in the UK Biobank.

“[...] we acknowledge that a current limitation of our work is that UK Biobank samples are limited in their diversity with individuals primarily falling between the ages of 40 to 69. In addition, the UK Biobank is known to contain racial and socioeconomic biases, which can lead to problematic inequities in terms of healthcare [3]. It would therefore be important to re-train or update our model on a more diverse population and perform a careful analysis of how well our model generalizes to underrepresented cohorts in future work before translating this method to hospital settings.”

9. MRI Data: only the 4 chamber long axis has been used. Was the reason for this due to technical reasons (as these data were available in all subjects at best quality) or due to a computation reason? Please comment on this and include a statement how to integrate further (more complex?) features.

We chose to use the long axis view of the cardiac MRI, since it is informative of the MRI-derived phenotypes considered in this work (e.g., LVM, RVEDV, etc.). Indeed, other views such as the short axis view can be informative for more general phenotypes such as BMI. To demonstrate this, we added an analysis, where we used our framework to integrate three modalities, short axis views, long axis views, and ECGs; see Fig. 3 above. In general, we observe that including the short axis view as a third modality improves the prediction of phenotypes such as BMI and sex, which are more accurately predicted from this view alone. We added Fig. 3 as Supplementary Fig. S2 in the revised manuscript together with the following sentence referencing the figure: “While we mainly apply our framework to integrate two modalities (ECGs and cardiac MRIs), we demonstrate that it can also be applied to three or more modalities in Supplementary Fig. S2.”

10. Were the results based on the multi-modal approach confirmed by a clinician, or was it confirmed by the ECG/MRI data itself? i.e. I assume that in some cases, the results were not 100% similar between MRI features and MRI-imputed features. How was this solved?

In our experiments, we used the ECG and cMRI measurements provided directly from the UK Biobank [1] for validating the generated ECGs and cMRIs. In particular, in Fig. 2 in our manuscript, we trained models to predict these provided measurements from the generated ECGs and cMRIs and then evaluated the performance of these predictions on a held out test set. We clarified this by adding the following sentence in the Results section: “All MRI- and ECG-derived phenotypes as well as the categorical and continuous-valued physiological phenotypes were provided by the UK Biobank.”

In addition, we evaluated the quality of the generated modalities by verifying that decoding translations in the latent space leads to interpretable shifts in the original modalities. For example, in Supplementary Fig. S10, we showed that generating ECGs from embeddings shifted along the latent space direction of increasing QT interval leads to a corresponding increase in QT interval in the generated ECG. Similarly, in Fig. 3 and Supplementary Fig. S10, we showed that the generated cMRIs accurately reflect changes upon increasing or decreasing LVM, RVEDV, and BMI.

References

1. Sudlow, C. *et al.* UK Biobank: An Open Access Resource for Identifying the Causes of a Wide Range of Complex Diseases of Middle and Old Age. *PLoS Medicine* **12** (2015).
2. Khurshid, S. *et al.* Deep Learning to Predict Cardiac Magnetic Resonance–Derived Left Ventricular Mass and Hypertrophy From 12-Lead ECGs. *Circulation: Cardiovascular Imaging* **14**, e012281 (2021).
3. Obermeyer, Z., Powers, B., Vogeli, C. & Mullainathan, S. Dissecting racial bias in an algorithm used to manage the health of populations. *Science* **366**, 447–453 (2019).
4. Radford, A. *et al.* Learning transferable visual models from natural language supervision. *arXiv:2103.00020* (2021).
5. Li, Y., Yang, M. & Zhang, Z. A survey of multi-view representation learning. *IEEE transactions on knowledge and data engineering* **31**, 1863–1883 (2018).

6. Hotelling, H. in *Breakthroughs in statistics* 162–190 (Springer, 1992).
7. Andrew, G., Arora, R., Bilmes, J. & Livescu, K. *Deep canonical correlation analysis* in *International Conference on Machine Learning* (2013).
8. Butler, A., Hoffman, P., Smibert, P., Papalexi, E. & Satija, R. Integrating single-cell transcriptomic data across different conditions, technologies, and species. *Nature Biotechnology* **36**, 411–420 (2018).
9. Stuart, T. *et al.* Comprehensive Integration of Single-Cell Data. *Cell* **177**, 1888–1902.e21 (2019).
10. Ning, Z., Xiao, Q., Feng, Q., Chen, W. & Zhang, Y. Relation-induced multi-modal shared representation learning for Alzheimer’s disease diagnosis. *IEEE Transactions on Medical Imaging* **40**, 1632–1645 (2021).
11. Zhou, T. *et al.* *Deep multi-modal latent representation learning for automated dementia diagnosis* in *International Conference on Medical Image Computing and Computer-Assisted Intervention* (2019), 629–638.
12. Shah, S. *et al.* Genome-wide association and Mendelian randomisation analysis provide insights into the pathogenesis of heart failure. *Nature Communications* **11** (2020).
13. Roselli, C. *et al.* Multi-ethnic genome-wide association study for atrial fibrillation. *Nature genetics* **50**, 1225–1233 (2018).
14. Van Setten, J. *et al.* Genome-wide association meta-analysis of 30,000 samples identifies seven novel loci for quantitative ECG traits. *European Journal of Human Genetics* **27** (Jan. 2019).
15. Smith, J. *et al.* Genome-wide association study of electrocardiographic conduction measures in an isolated founder population: Kosrae. *Heart rhythm : the official journal of the Heart Rhythm Society* **6**, 634–641 (2009).
16. Pirruccello, J. *et al.* Analysis of cardiac magnetic resonance imaging in 36,000 individuals yields genetic insights into dilated cardiomyopathy. *Nature Communications* **11**, 2254 (2020).
17. Petersen, S. E. *et al.* UK Biobank’s cardiovascular magnetic resonance protocol. *Journal of cardiovascular magnetic resonance* **18**, 1–7 (2015).

REVIEWERS' COMMENTS

Reviewer #1 (Remarks to the Author):

The authors have adequately addressed my comments with additional analyses and/or acknowledgment of limitations, as appropriate. I have no further comments.

Reviewer #3 (Remarks to the Author):

Thanks to the authors for providing a good reply to my comments and performance of additional analyses.

In my view the manuscript has been improved.

I recommend the follow two points to be added to the discussion:

- Inclusion of biomarkers/LDL: add the "unexpected" findings to lower LDL in subjects with hyperchol (maybe due to lipid-lowering medication) in the discussion. One possibility would be to adjust for lipid-lowering medication; I assume that these data are not available in the UKBB.

- Overlap of GWAS findings with "normal" GWAS Data. As shown in the Venn Diagrams there is a large proportion of SNPs that had previously not be identified (much larger than the number of SNPs previously been known but not identified by the new approach).

Add this to the discussion including possible explanations.

Reviewer #1 response:

The authors have adequately addressed my comments with additional analyses and/or acknowledgment of limitations, as appropriate. I have no further comments.

We are glad to have addressed the reviewer's comments.

Reviewer #3 response:

Reviewer Comments:

Thanks to the authors for providing a good reply to my comments and performance of additional analyses. In my view the manuscript has been improved.

We are glad to have addressed the reviewer's comments and have updated the manuscript according to their recommendations below.

I recommend the follow two points to be added to the discussion: - Inclusion of biomarkers/LDL: add the "unexpected" findings to lower LDL in subjects with hyperchol (maybe due to lipid-lowering medication) in the discussion. One possibility would be to adjust for lipid-lowering medication; I assume that these data are not available in the UKBB.

We have now added the following to the discussion regarding this point.

Added to Discussion: "For deployment in such settings, it is critical to account for potential confounding factors. For example, in our dataset, LDL was negatively correlated with incidence of hypercholesterolemia, which is presumably due to these individuals taking lipid lowering medications."

- Overlap of GWAS findings with "normal" GWAS Data. As shown in the Venn Diagrams there is a large proportion of SNPs that had previously not be identified (much larger than the number of SNPs previously been known but not identified by the new approach). Add this to the discussion including possible explanations.

Indeed, each traditional GWAS is only able to capture SNPs associated with a single phenotype. On the other hand, by working with modalities directly, our unsupervised GWAS is able to more broadly capture SNPs associated with the cardiovascular system. Thus, our method is able to capture the majority of SNPs found by traditional GWAS. We have now added the following to the discussion.

Added to Discussion: "Our unsupervised GWAS was able to leverage information across ECGs and cardiac MRIs to capture a wide range of SNPs that had an impact on the cardiovascular system. Subsets of these SNPs were previously found by traditional supervised approaches on individual phenotypes. Given that our approach is cross-modal, we also identified SNPs that had not been found by previous GWAS approaches. This novel framework for performing unsupervised GWAS in cross-modal representations also opens important avenues for future work. Investigating the differences in the identified SNPs between unsupervised and traditional GWAS is an interesting direction for future work and requires careful consideration of potential confounders. "